



# Optimal use of Prede POM sky radiometer for aerosol, water vapor, and ozone retrievals

Rei Kudo[1], Henri Diémoz[2], Victor Estellés[3,4], Monica Campanelli[4], Masahiro Momoi[5], Franco Marenco[6], Claire L. Ryder[7], Osamu Ijima[8], Akihiro Uchiyama[9], Kouichi Nakashima[10], Akihiro Yamazaki[1], Ryoji

Nagasawa[1], Nozomu Ohkawara[1], and Haruma Ishida[1]

[1]Meteorological Research Institute, Japan Meteorological Agency, Tsukuba, 305-0052, Japan

[2]ARPA Valle d'Aosta (Aosta Valley Regional Environmental Protection Agency), Saint-Christophe (Aosta), Italy

[3]Dept. Física de la Terra i Termodinàmica, Universitat de València, Burjassot, Valencia, Spain

[4]Consiglio Nazionale delle Ricerche, Istituto Scienze dell'Atmosfera e del Clima, via Fosso del Cavaliere, 100, 00133 - Roma, Italy

[5]Center for Environmental Remote Sensing, Chiba University, Chiba, 263-8522, Japan

[6]Space Applications and Nowcasting, Met Office, Exeter, EX1 3PB, UK

[7]Department of Meteorology, University of Reading, Reading, RG6 6BB, UK

[8]Aerological Observatory, Japan Meteorological Agency, Tsukuba, 305-0052, Japan

[9]National Institute for Environmental Studies, Tsukuba, 305-0053, Japan

[10]Japan Meteorological Agency, Tokyo, 100-8122, Japan

*Correspondence to*: Rei Kudo (reikudo@mri-jma.go.jp)

**Abstract.** The Prede POM sky radiometer is a filter radiometer deployed worldwide in the SKYNET international network.

A new method called, Skyrad pack MRI version 2 (MRI v2), is here presented, to retrieve aerosol properties (size distribution, real and imaginary parts of the refractive index, single-scattering albedo, asymmetry factor, lidar ratio, and linear depolarization ratio), and water vapor and ozone column concentrations from the sky radiometer measurements. MRI v2 overcomes two limitations of previous methods (Skyrad pack versions 4.2 and 5, and MRI version 1). One is the use of all the wavelengths of 315, 340, 380, 400, 500, 675, 870, 940, 1020, 1627, and 2200 nm, if available from the sky radiometers, for

example, in POM-02 models. The previous methods cannot use the wavelengths of 315, 940, 1627, and 2200 nm. This enables us to provide improved estimates of the aerosol optical properties, covering almost all the wavelengths of solar radiation. The other is the use of measurements in the principal plane geometry in addition to the solar almucantar plane geometry that is used in the previous versions. The measurements in the principal plane are regularly performed, however they are currently not exploited despite being useful in the case of small solar zenith angles, when the scattering angle distribution for almucantars

becomes too small to yield useful information. Moreover, in the inversion algorithm, MRI v2 optimizes the smoothness constraints of the spectral dependencies of the refractive index and size distribution,   and changes the contribution of the



diffuse radiances to the cost function according to the aerosol optical depth. These overcome issues with the estimation of the size distribution and single-scattering albedo in the Skyrad pack version 4.2. The scattering model used here allows for non-spherical particles, improving results for mineral dust and permitting to evaluate the depolarization ratio.

35       An assessment of the retrieval uncertainties using synthetic measurement show that best performance is obtained when the aerosol optical depths is larger than 0.2 at 500 nm. Improvements over the Skyrad pack versions 4.2 and 5 are obtained for the retrieved size distribution, imaginary part of the refractive index, single-scattering albedo, and lidar ratio at Tsukuba, Japan, while yielding comparable retrievals of the aerosol optical depth, real part of the refractive index, and asymmetry factor. A radiative closure study using surface solar irradiances from Baseline Surface Radiation Network and the

parameters retrieved from MRI v2 showed consistency, with a positive bias of the simulated global irradiance, about +24 $Wm^{-2}$ (+3 %). Furthermore, the MRI v2 retrievals of the refractive index, single-scattering albedo, asymmetry factor, and size distribution have been found in agreement with integrated profiles of aircraft in-situ measurements of two Saharan dust events at the Cape Verde archipelago, during the SAVEX-D 2015 field campaign.

## 1 Introduction

Aerosols, water vapor, and ozone are the most impacting factors on the atmospheric radiative budget in the solar wavelength band under cloudless sky. Indeed, the scattering and absorption of solar radiation by aerosols, as well as absorption by water vapor and ozone, have an important effect in the ultraviolet, visible, and near infrared wavelength regions. It is essential to observe these temporal and spatial changes and to evaluate their impacts on the atmospheric radiative budget and climate change (IPCC, 2013).

50       The columnar properties of aerosol, ozone and water vapor can be retrieved by ground-based remote sensing using sun-sky radiometers. A sun-sky radiometer is a narrow-band filter photometer that measures the solar direct radiation and the angular distribution of the diffuse radiation usually at ultraviolet, visible, and near infrared wavelengths. Such instruments are deployed worldwide in the international networks of AERONET (Holben et al., 1998) and SKYNET (Nakajima et al., 2020). Specifically, the Prede POM sky radiometer is the standard instrument from SKYNET, and now more than 100 instruments of

this kind are used in the world. Methods to retrieve aerosol properties, and water vapor and ozone column concentrations from the sky radiometer have been developed in the last 30 years. Nakajima et al. (1996) developed the "Skyrad pack", which is an all-in one package including methods for the calibration of the sky radiometer and for the retrieval of the aerosol physical and optical properties from the solar direct and diffuse radiation at the wavelengths of 340, 380, 400, 500, 675, 870, and 1020 nm. The products of the Skyrad pack version 4.2 (Skyrad v4.2) are the volume size distribution (VSD), real and imaginary parts of

the refractive index (RRI and IRI), aerosol optical depth (AOD), single scattering albedo (SSA), and phase function. The AOD (related to the columnar burden of aerosols), SSA (ratio of scattering to scattering + absorption), and phase function (angular distribution of scattering) or asymmetry factor (ASM; a measure of preferred direction of forward/backward scattering) are necessary to evaluate the impact of aerosols on the atmospheric radiative balance. Kobayashi et al. (2006) later developed the





Skyrad pack MRI (Meteorological Research Institute) version 1 (MRI v1) as a derivative of the Skyrad pack mainstream series.

MRI v1 is based on a statistical optimal estimation algorithm similar to the retrieval method employed within the NASA AERONET network (Dubovik and King, 2000). More recently, Kobayashi et al. (2010) introduced treatment for randomly oriented spheroidal particles in MRI v1, based on the data table developed by Dubovik et al. (2006). The phase function of dust particles estimated from spheroids is a more accurate representation than the spherical approximation used in previous versions of the software. Alongside, Hashimoto et al. (2012) upgraded the Skyrad pack version 4.2 to the version 5 (Skyrad

v5). They also introduced the statistical optimal estimation algorithm and a data quality control method. The products available from the Skyrad v5 and MRI v1 are similar to the ones that can be derived from the Skyrad v4.2.

In addition to this, the sky radiometer can measure the direct and diffuse radiation at 315 and 940 nm, which is absorbed by ozone and water vapor. Khatri et al. (2014) developed a calibration method from measurements at 315 nm and retrieved total ozone (TO3) from the direct irradiance. Uchiyama et al. (2018) calibrated the sky radiometer measurement at

940 nm by the Langley method from observations taken at a high mountain site. Campanelli et al. (2014, 2018) and Uchiyama et al. (2018) developed the calibration methods based on the modified Langley method (Reagan et al. 1986; Bruegge et al., 1992; Halthore et al., 1997) and showed the application to the sites other than high mountain. The modified Langley-based methods need the empirical equation to calculate the transmittance at 940 nm. Momoi et al. (2020) developed the on-site self-calibration method which does not require the empirical equation. They estimated the calibration constant at 940 nm by using

the dependency of the angular distribution of the diffuse radiances normalized to the direct irradiance on the PWV. All the methods showed the PWV was successfully retrieved from the calibrated sky radiometer measurement at 940 nm.

SKYNET has collaborated with international lidar networks, such as AD-Net (Sugimoto et al., 2015). In the frame of these activities, a synergistic method, SKYLIDAR, was developed to estimate the vertical profiles of extinction coefficient, SSA, and ASM from the sky radiometer and lidar measurements (Kudo et al., 2016). This enabled us to estimate the

atmospheric heating rate by remote sensing techniques (Kudo et al., 2016; 2018). Another synergistic approach employs the particle extinction-to-backscatter ratio (lidar ratio; LIR) and the linear depolarization ratio (DEP). These are important aerosol optical properties observed by Raman lidars and High-Spectral-Resolution Lidars (HSRL) and have been used for the aerosol typing (e.g., Burton et al., 2012; Groß et al., 2015). Recently, the LIR and DEP are included in the version 3 of the AERONET products, and some aerosol typing studies have been already conducted (e.g., Shin et al. 2018). The relations between LIR,

DEP, and aerosol types based on the lidar observations are utilized in these studies. Conversely, the LIR derived from sun-sky radiometer can be utilized instead of an assumed value to estimate the vertical profile of the extinction coefficient from conventional elastic backscatter lidars, which are deployed worldwide.

In this study, we developed a new method, the Skyrad pack MRI version 2 (MRI v2), to retrieve aerosol properties (VSD, RRI, IRI, AOD, SSA, ASM, LIR, and DEP), PWV, and TO3 from the sky radiometer data. Our method has two

advantages, compared to Skyrad v4.2, v5, and MRI v1. Firstly, MRI v2 is able to use the observations at all the available wavelengths from the sky radiometer, from 315 to 2200 nm, and simultaneously retrieves the aerosol optical properties, the PWV, and the TO3. This possibility was not available in Skyrad v4.2, v5, and MRI v1 where only the following wavelengths



were exploited: 340, 380, 400, 500, 675, 870, and 1020 nm. Since the retrieved aerosol optical properties cover a good part the solar wavelength region from 300 to 3000 nm, a detailed characterization of the radiative transfer in the short wavelength

under clear sky conditions is thus available from the sky radiometer measurements. Secondly, our method can be applied to both scanning patterns of the sky radiometer, i.e. solar almucantar and principal plane geometries. The preferred and most used scanning pattern is the almucantar geometry, but principal plane measurements are useful in the case of the small solar zenith angles because in that case the range of scattering angles obtained with the almucantar geometry is too small. Skyrad pack versions earlier than v3 allowed the users to analyze the scanning data in the principal plane geometries. However, the recent

retrieval methods of Skyrad v4.2, v5, and MRI v1 could only be applied to the data obtained from the almucantar geometry, which prevented the analysis of the data routinely collected in the principal plane geometry. This reduced the amount of observations available particularly at the observational sites at the low latitudes.

The sky radiometer data used in this study is described in Sect. 2. The algorithms of the MRI v2 retrieval method and the simulation of the surface solar irradiance using the MRI v2 retrieved parameters are described in Sect. 3. The retrieval

uncertainty is evaluated using the simulated data of the sky radiometer in Sect. 4. In Sect. 5, the results of the application of MRI v2 to the measurements at Tsukuba, Japan, and at Praia, Cape Verde are shown. The MRI v2 products are compared with the Skyrad v4.2 and v5 products, and the aircraft in-situ measurements. All the results are summarized in section 6.

## 2 Data

### 2.1 Observation at Tsukuba, Japan

Our newly developed method was applied to the measurements of the sky radiometer model POM-02 (Prede Co., Ltd., Tokyo, Japan) from February to October in 2018 at the Meteorological Research Institute, Japan Meteorological Agency (36.05˚N, 140.13˚E, about 25 m a.s.l.) in Tsukuba, Japan, about 50 km northeast of Tokyo. This instrument measures the solar direct irradiance and the angular distribution of the diffuse irradiances at the scattering angles of 2, 3, 4, 5, 7, 10, 15, 20, 25, 30, 40, 50, 60, 70, 80, 90, 100, 110, 120, 130, 140, 150, and 160˚ in the solar almucantar (ALM) or principal plane (PPL) geometries.

The measurable maximum scattering angle depends on the solar zenith angle ($\theta_0$) and is $2\theta_0$ for ALM geometry. The measurable scattering angle for PPL geometry is $\theta_0 + 60°$ due to the motion range of the sky radiometer. For the comparison of the retrievals from the diffuse radiances in the ALM and PPL geometries, we used a different observation schedule compared to the SKYNET standard one. The latter performs the scanning in the ALM geometry at every 10 minutes, while scanning in the PPL geometry is conducted in only the case that the solar zenith angle is less than 15˚. Our procedure performs a scanning

in the ALM and PPL geometries at every 15 minutes, regardless the value of the solar zenith angle. The measured wavelengths are 315, 340, 380, 400, 500, 675, 870, 940, 1020, 1627, and 2200 nm, and their full width at half maximum is 3±0.6 nm for near ultraviolet wavelengths, 10±2.0 nm for visible wavelengths, and 20±4.0 nm for near infrared wavelengths, respectively (Uchiyama et al., 2018a).


Our retrieval method uses the atmospheric transmittances ($T_d$) and the diffuse radiance normalized by the direct

irradiances ($R$). $T_d$ is obtained from the direct irradiance measurement ($V_d$) by giving the calibration constant ($F_o$):

$$T_d(\lambda) = \frac{R_{es}^2 V_d(\lambda)}{F_o(\lambda)} = \exp\left[-m_o\left(\tau_R(\lambda) + \tau_A(\lambda) + \tau_G(\lambda)\right)\right], \qquad (1)$$

$$m_o = 1/\cos\theta_o \qquad (2)$$

where $\lambda$ is the wavelength, $R_{es}$ is the sun-Earth distance in astronomical units, $m_o$ is the optical air mass, and $\tau_R$, $\tau_A$, and $\tau_G$

are the optical depths of Rayleigh scattering, aerosol extinction, and gas absorption, respectively. $R$ is calculated by

$$R(\Theta, \lambda) = \frac{V_s(\Theta, \lambda)}{V_d(\lambda) m_o \Delta\Omega(\lambda)} \qquad (3)$$

where $V_s$ is the diffuse irradiance measurement, $\Theta$ is the scattering angle, and $\Delta\Omega$ is the solid view angle. The solid view angle

is determined by scanning the distribution of radiation around the solar disk (Nakajima et al., 1996; Uchiyama et al., 2018b;

Nakajima et al. 2020). The diffuse radiance measurement is described as $V_s(\Theta, \lambda)/\Delta\Omega(\lambda)$, and the multiplication by $1/V_d(\lambda)$

cancels the calibration constant included in $V_s(\lambda)$ and $V_d(\lambda)$ because the direct irradiance and diffuse radiance are measured

by the same sensor. Only diffuse radiance at scattering angles larger than 3˚ are used in MRI v2, since at the scattering angle

of 2˚ abnormally large values were seen in the data. In addition, the diffuse radiances at 1627 and 2200 nm at the scattering

angles higher than 30˚ were also removed because of the weak scattering of solar radiation and low sensitivity of the detector

at 1627 and 2200 nm (Uchiyama et al., 2019).

The calibration constants at 340, 380, 400, 500, 675, 870, 940, 1020, 1627, and 2200 nm were transferred from our

reference sky radiometer by side-by-side comparison. The reference sky radiometer was calibrated by the Langley method

using the observation data at the NOAA Mauna Loa observatory in Hawaii, U. S. A.; 19.54˚N, 155.58˚W, 3397.0 m a.s.l.

(Uchiyama et al., 2014; 2018a). The calibration constant at 315 nm was determined by accounting for the TO3 measured by

the Brewer spectrophotometer at the Aerological Observatory, Japan Meteorological Agency, located next to the

Meteorological Research Institute. The calibration procedure of 315 nm is described in Appendix A.

Completely clear sky conditions are required for accurate retrievals. Therefore, in Sect. 5.1, we selected the clear

sky conditions based on the method by Kudo et al. (2010). The method judges the clear sky condition from the temporal

variations of the surface solar irradiance measured by a co-located pyranometer.

The measurements of the surface solar irradiances at the Aerological Observatory were used for verifying the

simulated surface solar irradiances using the retrieved aerosol properties, PWV, and TO3. The Aerological Observatory is a

station of BSRN (Baseline Surface Radiation Network; Driemel et al., 2018). The solar direct and hemispheric diffuse

irradiances are measured by pyrheliometer (Kipp & Zonen CHP21), and pyranometer (Kipp & Zonen CMP22) with a shading

cube in front of the sun. The global irradiances are obtained by the sum of the direct and hemispheric diffuse irradiance

measurements. The pyrheliometer and pyranometer are calibrated every 5 years by Japan Meteorological Agency and traceable

to the WRR (World Radiometric Reference). The BSRN measurement errors are 2 % for global, 0.5 % for direct, and 2 % for

diffuse irradiance (McArthur, 2005).



## 2.2 SAVEX-D

The Sunphotometer Airborne Validation Experiment in Dust (SAVEX-D) was conducted in August 2015 in the Cape Verde archipelago (Estellés et al., 2018), in conjunction with two airborne campaigns: AERosol properties – Dust (AER-D) and Ice in Clouds Experiment – Dust (ICE-D) over the eastern tropical Atlantic. The main objective of the SAVEX-D was the

validation of the SKYNET and AERONET aerosol products in conditions dominated by Saharan dust, with aircraft in-situ measurements performed and integrated in the vertical. Two flights were successfully carried out under clear sky conditions on 16 and 25 August near Praia (14.948°N, 23.483°W, 128 m ASL) and Sal (16.733°N, 22.935°W, 60 m ASL) islands in the Cape Verde archipelago, respectively. The Saharan dust originating from Africa was observed during the two flights with AOD at 500 nm higher than 0.5 and 0.2, respectively. More details from the field campaigns were made available by Marenco

et al. (2018) and Ryder et al. (2018).

A sky radiometer model POM-01 was deployed at Praia airport during SAVEX-D. We applied Skyrad MRI v2 to the sky radiometer data of the solar direct irradiances and the diffuse radiances in the ALM geometry at the wavelengths of 443, 500, 675, 870, 1020 nm (Estellés et al., 2018). Note that even measurements at a non-standard wavelength of 443 nm can be processed by our algorithm, since wavelengths used in our retrieval method can be flexibly customized to the measurements.

In this study, the VSD, RRI, IRI, SSA, and ASM of the MRI v2 products were compared with those derived from the in-situ measurements (Ryder et al., 2018). The details of the aircraft in-situ measurements and methods to derive the aerosol physical and optical properties were described in Ryder et al. (2018).

## 3 Algorithms

### 3.1 Retrieval of aerosols, precipitable water vapor, and total ozone

#### 3.1.1 Inversion strategy

Our retrieval method is based on an optimal estimation technique similar to the one employed in the AERONET retrieval (Dubovik and King, 2000). The VSD, RRI, IRI, PWV, and TO3 are simultaneously optimized to all the measurements of the sky radiometer and all the a priori constraints. The best solution is obtained by minimizing the objective function,

$$f(\mathbf{x}) = \left(\mathbf{y}^{obs} - \mathbf{y}(\mathbf{x})\right)^T (\mathbf{W}^2)^{-1}\left(\mathbf{y}^{obs} - \mathbf{y}(\mathbf{x})\right) + \mathbf{y}_a(\mathbf{x})^T (\mathbf{W}_a^2)^{-1}\mathbf{y}_a(\mathbf{x}),  \tag{4}$$

where $\mathbf{x}$ is a state vector to be optimized, the vector $\mathbf{y}^{obs}$ represents measurements, the vector $\mathbf{y}(\mathbf{x})$ represents the simulations by the forward model corresponding to $\mathbf{y}^{obs}$, $\mathbf{W}^2$ is the covariance matrix of $\mathbf{y}$, the vector $\mathbf{y}_a(\mathbf{x})$ is an a priori constraint for $\mathbf{x}$, and $\mathbf{W}_a$ is an associated covariance matrix. The minimization of $f(\mathbf{x})$ is conducted with the algorithm developed by Kudo et al. (2016). A logarithmic transformation is applied to $\mathbf{x}$ and $\mathbf{y}$. The minimum of $f(\mathbf{x})$ in the log $(\mathbf{x})$-space is searched by the iteration of $\mathbf{x}_{i+1} = \mathbf{x}_i + \alpha\mathbf{d}$, where vector $\mathbf{d}$ is determined by the Gauss-Newton method, and a scalar $\alpha$ is determined by the

line search with Armijo rule.



### 3.1.2 Measurement and state vectors

$\mathbf{y}^{obs}$ comprises the transmittances at the wavelengths of 315, 340, 380, 400, 500, 675, 870, 940, 1020, 1627, and 2200nm, and the normalized diffuse radiances at scattering angles larger than 3˚ in the ALM or PPL geometries. Note that the wavelengths and scattering angles used in our method can be arbitrarily selected. For example, we used the wavelengths of 443, 500, 675,

870, 1020 nm in Sect. 5.2.

Similarly to the retrieval methods of Dubovik and King (2000) and Kobayashi et al. (2010), the covariance matrix $\mathbf{W}^2$ of Eq. (4) was assumed to be diagonal, and their values were given by the measurement errors of the transmittance and normalized diffuse radiance. The measurement error of the transmittance mainly depends on the uncertainty of the calibration constant. Uchiyama et al. (2018a) estimate the error of the calibration constant determined by the Langley method using the

observation data at the NOAA Mauna Loa Observatory to be from 0.2 to 1.3 %, and the error due to the transfer of the calibration constant from the reference instrument by the side-by-side comparison was from 0.1 to 0.5 %. Therefore, we gave the value of 2 % as the measurement errors of the transmittances at all the wavelengths. The measurement errors of normalized diffuse radiances are defined as 5 % in the work of Kobayashi et al. (2006). We also employed the same value, but we introduced a dynamic weight factor depending on the AOD as follows,

$$W = \min\left\{5\% * \max\left[\left(\frac{0.3}{\tau_A(\lambda)}\right)^2, 1.0\right], \ 100\%\right\}, \tag{5}$$

where $\tau_A(\lambda)$ is the AOD at wavelength $\lambda$. This factor increases with a decrease of AOD and takes into account the fact that the absolute value of the diffuse radiance, as well as the signal-to-noise ratio, decrease with decreasing AOD. In actual measurements, the angular distribution of the diffuse radiances at 1627 and 2200 nm has unnatural oscillations with scattering angles in the cases of low AOD. When $\tau_A(\lambda)$ is more than 0.3, the value of $W$ is 5 %. The value of 0.3 was empirically

determined by many trials of applying the different values of $\tau_A(\lambda)$ to the measurements at Tsukuba.

The components of $\mathbf{x}$ are the PWV, TO3, RRI and IRI at 315, 340, 380, 400, 500, 675, 870, 940, 1020, 1627, and 2200nm, and the parameters describing VSD. We assumed that the VSD consists of spherical and non-spherical particles and is expressed as the combination of 20 lognormal distributions in the range of the particle radius from 0.03 to 30.0 μm:

$$\frac{dV(r)}{d\ln r} = \sum_{i=1}^{20} C_i \exp\left[-\frac{1}{2}\left(\frac{\ln r - \ln r_{m,i}}{s_i}\right)\right] \tag{6}$$

$$= \sum_{i=1}^{20}\left\{\varepsilon_i C_i \exp\left[-\frac{1}{2}\left(\frac{\ln r - \ln r_{m,i}}{s_i}\right)\right] + (1-\varepsilon_i)C_i \exp\left[-\frac{1}{2}\left(\frac{\ln r - \ln r_{m,i}}{s_i}\right)\right]\right\}, \tag{7}$$

$$\varepsilon_i = \begin{cases}1, & r_i < r_{lm} \\ \varepsilon, & r_i \geq r_{lm}\end{cases}, \tag{8}$$

where $r$ is particle radius, $V(r)$ is volume, $C_i$, $r_{m,i}$, $s_i$, and $\varepsilon_i$ are maximum volume, center radius, width, and volume ratio of the spherical particle to the sum of spherical and non-spherical particles for each lognormal distribution, respectively. The first term of Eq. (7) refers to spherical particles, and the second term to non-spherical particles. $r_{lm}$ is a radius to separate $\frac{dV(r)}{d\ln r}$ into

the fine and coarse modes. It is defined as the radius at the local minimum of the $\frac{dV(r)}{d\ln r}$ and is determined at every iterative



steps of $\mathbf{x}_{i+1} = \mathbf{x}_i + \alpha \mathbf{d}$ in the minimization process of $f(\mathbf{x})$. We assumed that the fine mode comprises only spherical particles, and the coarse mode is a mixture of non-spherical and spherical particles with a ratio of $\varepsilon$. The optimized parameters of the size distribution are $C_i$, and $\varepsilon$. $r_{m,i}$ is fixed by the radius which separate the range of 0.03 and 30 μm at log-spaced intervals, $\ln\Delta r$. The $s_i$ is also fixed by $\ln\Delta r/1.65 \approx 0.21$. The value of 1.65 is empirically selected from the range of $s_i$, which satisfies the

following two conditions. The first condition determines the maximum value of $s_i$. The observed width of the fine mode of the VSD is smaller than that of the coarse mode and is about 0.4 (Dubovik et al., 2002). Since we express the VSD as the combinations of the lognormal distributions (Eq. 6), the $s_i$ should be smaller than 0.4. The second condition is the minimum value of $s_i$. The $\frac{dV(r)}{d\ln r}$ at the middle radius of two lognormal distributions, $\ln(r) = 0.5\left(\ln(r_{m,i}) + \ln(r_{m,i+1})\right)$, should be larger than $0.5(C_i + C_{i+1})$. If not so, the shape of $\frac{dV(r)}{d\ln r}$ has unnatural oscillations. Hence, the $s_i$ should be larger than

$\ln\Delta r/2.35$, where $2.35 s_i$ is the full width at half maximum of the lognormal distribution.

The admitted radii of the VSD range from 0.03 to 30 μm in MRI v2. However, the radius range of the previous SKYNET retrieval methods is from 0.01 to 20.0 μm. The radius range used in the AERONET retrieval is from 0.05 to 15 μm. We investigated the radius range which can be actually estimated from all the sky radiometer data by a similar technique as in Tonna et al. (1995). Figure A2 shows the Mie kernel functions of scattering and extinction for wavelengths from 315 to 2200

nm and scattering angles from 2° to 120°. We can see that the sky radiometer measurements carry information of the VSD in the radius range from 0.02 μm to 30 μm approximately. When we limit the scattering angle range from 3° to 60° and the wavelength range from 340 to 1020 nm, the radius range that can be retrieved is roughly from 0.03 μm to 10 μm. We might retrieve the giant particles around 20 μm from the diffuse radiances at the wavelengths of 1627 and 2200 nm at a scattering angle of 2°. In practice, the measurement errors of the diffuse radiances at the scattering angle of 2° is too large for almost all

wavelengths. However, we can use the measurements at 1627 and 2200 nm at a scattering angle of 3°. Therefore, compared with the previous retrieval methods, we extended the retrieval range of the size distribution to the radius of 30 μm.

### 3.1.3 Forward modeling

The forward model $\mathbf{y}(\mathbf{x})$ calculates the transmittances and the normalized diffuse radiances from $\mathbf{x}$. The aerosol extinction and scattering coefficients, and phase function for the spherical particle are calculated by the Mie theory. For the non-spherical

particles, we employed the optical properties of the randomly oriented spheroids with a fixed aspect ratio distribution, which is optimized to the laboratory measurement of mineral dust (feldspar sample) phase matrices (Dubovik et al., 2006). The vertical profile of aerosols can be customized, but at this first stage it is assumed to be constant from the surface to the altitude of 2 km.

The gaseous absorption coefficients for water vapor and ozone are calculated by the correlated k-distribution (CKD)

method according to the inputs of the PWV and TO3. The data table of the CKD method is developed by Sekiguchi and Nakajima (2008) using the HITRAN 2004 database. The vertical profile of ozone is given from the 1976 version of the U.S. standard atmosphere. The vertical profiles of water vapor, temperature, and pressure are also given from the U.S. 1976 standard





atmosphere, but optionally we can select other auxiliary data. For example, we used the data of the radiosonde launched at the Aerological Observatory in Sect. 5.1, while the U.S. 1976 standard atmosphere was used in Sect. 5.2. Other than the water

vapor and ozone, the gaseous absorption of $CO_2$, $N_2O$, $CO$, $CH_4$, and $O_2$ are considered in the forward model. Their vertical profiles were given from the standard atmosphere, and their absorption coefficients were calculated by the CKD method.

The solar direct irradiances and the diffuse radiances in the ALM and PPL geometries are calculated by the radiative transfer model, RSTAR (Nakajima and Tanaka, 1986; 1988). The diffuse radiances were calculated using the IMS method (Nakajima and Tanaka, 1988), which is an approximation method to simulate the diffuse radiances near the sun. For only the

calculation at wavelengths of 315, 940, and 2200 nm, the response function of the interference filter of the sky radiometer was taken into account because the spectral changes in the absorption of ozone and water vapor within the filter band width cannot be ignored. We divided the 315 and 940 nm bands into five sub-bands, and the 2200 nm band into three sub-bands. For other wavelengths, the monochromatic calculation was assumed. We also incorporated the vector radiative transfer model, PSTAR (Ota et al., 2010) as an alternative to the scalar model RSTAR. The two radiative transfer codes can be easily switched. The

option of the PSTAR will be useful if polarization measurements, such as in AERONET (Holben et al., 1998), are introduced to the sky radiometer in the future. Furthermore, we parallelized the codes of RSTAR and PSTAR using "OpenMP" because radiative transfer calculations at the wavelengths from 315 to 2200 nm are time-consuming.

### 3.1.4 A priori constraints

A smoothness constraint for the refractive index and size distribution is necessary for a stable retrieval (Dubovik and King,

2000). We constraint the spectral dependencies of the RRI and IRI by limiting the values of the following first derivatives of the refractive index with the wavelength,

$$\mathbf{y_a}(\mathbf{x}) = \left( \cdots \quad \frac{\ln(n(\lambda_i)) - \ln(n(\lambda_{i+1}))}{\ln(\lambda_i) - \ln(\lambda_{i+1})} \quad \cdots \quad \frac{\ln(k(\lambda_i)) - \ln(k(\lambda_{i+1}))}{\ln(\lambda_i) - \ln(\lambda_{i+1})} \quad \cdots \right), \tag{9}$$

$(i = 1, \cdots N_\lambda\text{-}1)$,

where $n$ and $k$ are the RRI and IRI at the wavelength $\lambda$, and $N_\lambda$ is the number of wavelengths. For the VSD, the second

derivatives of $C_i$ (Eq. 6) with respect to the particle radius is introduced by,

$$\mathbf{y_a}(\mathbf{x}) = \left( \cdots \quad \ln(C_{i-1}) - 2\ln(C_i) + \ln(C_{i+1}) \quad \cdots \right), \tag{10}$$

$(i = 1, \cdots 20)$, $C_0 = 0.1 \times C_1^0$, $C_{21} = 0.1 \times C_{20}^0$,

where $C_0$ and $C_{21}$ are the volumes outside the radius range from 0.03 to 30.0 μm, and $C_1^0$ and $C_{20}^0$ are the initial values of $C_i$ in the iteration of the Gauss-Newton method (Sect. 3.1.5). The small values of $C_0$ and $C_{21}$ prevent $C_1$ and $C_{20}$ from being

abnormal values. The denominator of the second derivative was ignored because the $r_{m,i}$ has an equal interval.

The covariance matrix $\mathbf{W_a^2}$ in Eq. (4) determines the strength of the smoothness constraints. We assumed that the matrix is diagonal, and the values of each element corresponding to the RRI, IRI, and VSD are set empirically. The typical ranges of the RRI and IRI for the tropospheric aerosols are from 1.4 to 1.6, and from 0.005 to 0.05, respectively, at the visible and near infrared wavelengths (Dubovik and King, 2000). We therefore defined the values of $\mathbf{W_a^2}$ as,





$W_a = \frac{\ln(1.6)-\ln(1.4)}{\ln(2200)-\ln(315)} \cong 0.07$ for RRI, (11)

$W_a = \frac{\ln(0.05)-\ln(0.005)}{\ln(2200)-\ln(315)} \cong 1.2$ for IRI. (12)

The typical VSD is expressed by a bi-modal lognormal distribution. The AERONET retrievals obtained in different aerosol conditions in the world (Dubovik et al., 2002) show that the width of the lognormal distribution for the fine mode is about half of that for the coarse mode. This suggests that the second derivative of the fine mode with respect to the particle

radius is also larger than that of the coarse mode. Therefore, different values of $W_a$ were given to the fine and coarse modes:

$W_a = \begin{cases} 1.6, & r_i < r_{lm} \\ 0.6, & r_i \geq r_{lm} \end{cases}$. (13)

These values were empirically determined based on the work of Dubovik and King (2000) and by numerous trial and error using the measurements of the SAVEX-D campaign.

### 3.1.5 Initial values and outputs

The objective function (Eq. 4) is minimized by the iteration of the Gauss-Newton method. The iterative method requires the initial value of **x**. The initial value of the RRI and IRI index is given as $1.50-0.005i$ at all the wavelengths. The ratio of the spherical particles in the coarse mode, $\varepsilon$, is 0.1. The volume of each lognormal distribution, $C_i$, is given from the size distribution created by the following procedure,

(1) AOD at weak gas absorption wavelengths, 340, 380, 400, 500, 675, 870, and 1020 nm are directly calculated from the

direct irradiances.

(2) Consider a bi-modal size distribution with fixed mode radii, 0.1 and 1.0 μm, and widths, 0.4 and 0.8, for the fine and coarse modes, respectively.

(3) Volume ratio between fine and coarse modes is fitted to Ångström exponent obtained from the AOD of (1).

(4) Total volume of the size distribution is fitted to the AOD at 500 nm.

After finding the best solution of **x**, the VSD, RRI, IRI, AOD, SSA, ASM, and phase function are provided as an output. In addition, we calculate the LIR and DEP because these are important optical properties in synergistic analyses using both the sky radiometer and lidar observations.

The objective function of Eq. (4) is a measure how much the **x** is optimized to the $\mathbf{y}^{obs}$. However, the objective function includes the terms of the a priori constraints and does not imply a fitting to only $\mathbf{y}^{obs}$. Therefore, we output another

measure of the fitness,

$f_{obs}(\mathbf{x}) = \sqrt{\frac{\left(\mathbf{y}^{obs}-\mathbf{y}(\mathbf{x})\right)^T (\mathbf{W}^2)^{-1} \left(\mathbf{y}^{obs}-\mathbf{y}(\mathbf{x})\right)}{N_y}},$ (14)

where $N_y$ is the number of the elements in the vector $\mathbf{y}^{obs}$. Eq. (14) is the mean of the differences between the sky radiometer measurements and ones calculated from the **x** by the forward model, weighted by their respective experimental uncertainties.



We can filter out the retrievals, which are not well optimized to the measurements, by giving a threshold to $f_{obs}(\mathbf{x})$. We used

the threshold of 1.0, and the retrieval results, which did not satisfy the condition of $f_{obs}(\mathbf{x}) > 1.0$, were discarded in this study. This means that almost all the elements of the vector $\mathbf{y}(\mathbf{x})$ lies in the range of $\mathbf{y}^{obs} \pm \mathbf{W}$.

### 3.2 Surface solar irradiance

In the study of the aerosol-radiation interaction, it is important to ensure the consistency between the observed and simulated surface irradiances. For this radiative closure study, the global, direct, and diffuse components of the surface solar irradiance

in the wavelength region from 300 to 3000 nm were calculated from the retrieved aerosol optical properties, PWV, and TO3, and we compared them with those observed at the Aerological Observatory in Sect. 5.1.

The surface solar irradiances were calculated by our developed radiative transfer model (Asano and Shiobara, 1989; Nishizawa et al. 2004; Kudo et al. 2011). Note that this model is different from RSTAR and PSTAR used in the forward model of MRI v2. The solar spectrum between 300 and 3000 nm was divided into 54 intervals. Gaseous absorption by water vapor,

carbon dioxide, oxygen, and ozone were calculated by the CKD method. The inputs to the radiative transfer model are AOD, SSA, and phase function at 54 wavelengths from 300 to 3000 nm. These were calculated from the retrieved VSD, RRI, and IRI. The RRI and IRI at wavelengths between 315 and 2200 nm were interpolated from the retrieved RRI and IRI in the log-log space. For the wavelengths less than 315 nm and more than 2200 nm, the retrieved RRI and IRI at 315 and 2200 nm were used. A main advantage of MRI v2 is that the aerosol optical properties are retrieved in a wavelength range almost covering

the whole short-wave band.

### 4 Uncertainties in retrieval products

### 4.1 Radiometric uncertainties

The uncertainties of the MRI v2 retrieval products were evaluated using the simulations of the sky radiometer measurements. The simulation was conducted for the three aerosol models of water-soluble, dust, and biomass burning (Table 1) used in the

accuracy assessment of the AERONET retrieval (Dubovik et al., 2000). In the simulation, normally distributed random errors were added to direct irradiances, diffuse radiances, and surface albedo. The standard deviations used in generating the random errors are described in Table 2. The AOD, solar zenith angle, PWV, and TO3 used in the simulation were randomly selected from the ranges in Table 1. We conducted 200 simulations for each of three aerosol models and two scanning patterns of the ALM and PPL geometries, respectively. Our retrieval method was applied to total 1200 simulation data sets. In 98 out of the

1200 results, $f_{obs}(\mathbf{x})$ was more than the threshold of 1.0. When the perturbations in the simulation data were too large, our retrieval method was not able to optimize the parameters to the simulation data. The 98 retrievals were not included in the following results.

Figure 1 shows the dependencies of the retrieval errors of SSA on the solar zenith angle and AOD. In this study, we define the retrieval error as a deviation of the retrieved value from the simulated value for each individual simulation. When





the solar zenith angle is small, the range of scattering angle for the diffuse radiance measurements is small, and the available information of the phase function from the diffuse radiances becomes smaller. We expected that this would affect the retrieval of aerosols, but no clear dependency of the retrieval errors of the SSA on the solar zenith angle was seen in Fig. 1. The retrieval errors of the other aerosol physical and optical properties, PWV, and TO3 also did not show any apparent dependencies on the solar zenith angle.

Small AODs make it difficult to retrieve the refractive index, because the diffuse radiances are less sensitive to the refractive index in this case (Dubovik et al., 2000). The retrieval errors of the SSA were obviously greater when the AOD was smaller than 0.2 (Fig. 1). This dependence on AOD was also seen in the retrieval errors of the other aerosol physical and optical properties.

The means (bias) and standard deviations (uncertainty) for the retrieval errors of the aerosol physical and optical
properties, PWV, and TO3 were summarized in Table 3. Note that the AOD in Table 3 is calculated from the retrieved VSD, RRI, and IRI and is not directly obtained from the direct solar irradiance. Overall, both the biases and uncertainties of the retrieval errors were large in the case that the AOD is less than 0.2. In particular, we note a positive bias in the IRI, and an uncertainty of more than 100 %. This bias affected the SSA and LIR, which depend on the IRI. The SSA showed a negative bias and the LIR a positive one. Even if the AOD at 500 nm was more than 0.2, the retrievals at the near infrared wavelengths
were biased in the results of the IRI, SSA, and LIR of the water-soluble and biomass burning models (bias +179 % and +59 %, respectively, with an uncertainty of ±400−450 %). This is because the AOD of the two aerosol models is low at near infrared wavelengths. Conversely, the retrieval errors at the near infrared wavelengths for the dust model were smaller (bias + 3 %, uncertainty 30 %). The retrieval errors of the DEP for the biomass burning also were more than 100 %. The reason of the large retrieval error is that the simulated value of the DEP is near zero.

Figure 2 illustrates the retrieved VSD for three aerosol models. The VSD is normalized to the total volume. The shaded area shows the radius range around the fine and coarse mode peaks, (mode radius ± one standard deviation). The bias and uncertainty of the VSD in Table 3 is calculated from the results in the shaded areas. The biases of the retrieval errors around the fine and coarse mode peaks were less than 22 % (Table 3), but the uncertainties were not small (Fig. 2). The retrieval errors were also large in the outside of the shaded areas (Fig. 2). In the result of the ALM geometry for dust model, there is a
single subset with the high fine mode and low coarse mode. Since the AOD at 500 nm used in this simulation was too small, 0.26E-4, the retrieval failed.

Overall, the retrieval results at the near ultraviolet and visible wavelengths were good in the case that the AOD at 500 nm was larger than 0.2: the absolute values of biases + uncertainties in the retrieval products were less than 0.04 for AOD, less than 0.05 for RRI, less than 130 % for IRI, less than 50 % for VSD, less than 0.05 for SSA, less than 0.02 for ASM, less
than 20 for LIR, and less than 60 % for DEP. Regardless of the AOD, the retrieval errors of the PWV and TO3 were less than 8 mm and 42 m atm-cm, respectively.



### 4.2 Uncertainty in the aerosol vertical profile

In the previous numerical experiments, we investigated the differences of the retrieval errors between the ALM and PPL geometries without finding clear differences. One reason is that, so far, we did not consider the error from the aerosol vertical

profile in the retrieval. The aerosol vertical profile affects the diffuse radiances in the PPL geometry (Torres et al., 2013; Momoi et al., 2020). Therefore, we now investigate the impacts of the aerosol vertical profile on the retrievals for the ALM and PPL geometries. To this purpose, we simulated the sky radiometer data for the dust and biomass burning models with different aerosol vertical profiles and conducted the retrieval with a fixed aerosol vertical profile from 0 to 2 km. Three patterns of the aerosol vertical profiles, constant from 0 to 2 km (P1), from 2 to 4 km (P2), and from 4 to 6 km (P3), were used in the

simulation. The constant profile makes it easy to understand the influences of the aerosol layer altitude. Other parameters were set as follows: the aerosol optical depth at 500 nm was 0.5, the PWV was 30 mm, the TO3 was 350 m atm-cm, and the solar zenith angle was 45˚.

Table 4 summarizes the means for the retrieval errors of all the parameters, and Figure 3 shows the retrieval errors of SSA and ASM. The impacts of the aerosol vertical profile on the retrieval errors for the ALM geometry were small, but the

retrieval error for the PPL geometry obviously depended on the aerosol vertical profile, even for the PWV and TO3. The retrieval errors of SSA for both the dust and biomass burning models were positive at the wavelengths smaller than 870 nm and negative at the wavelengths larger than 870 nm (Fig. 3). These positive and negative errors were related to the diffuse radiances. Figure 4 shows the mean ratio of the simulated diffuse radiances to those of P1 over the scattering angles. The mean ratios of P2 and P3 for the PPL geometry were more than 1.0 at the wavelengths smaller than 870 nm. Only the mean ratios at

940 and 2200 nm, which have strong water vapor absorption, were less than 1.0. The mean ratios at 1020 and 1627 nm were about 1.0. The strong diffuse radiances of P2 and P3 at the wavelengths smaller than 870 nm for the PPL geometry caused the underestimation of the IRI (Table 4) and the overestimation of the SSA (Fig. 3 and Table 4). The weak diffuse radiances of P2 and P3 at 940 and 2200 nm for the PPL geometry increased the IRI and decreased the SSA. The changes in the diffuse radiances at 1020 and 1627 nm is negligible but the SSA at these wavelengths were underestimated (Fig. 3). This is because the combined

effect of the increased IRI at 940 and 2200 nm and the smoothness constraint of the spectral change for the IRI increased the IRI at 1020 and 1627 nm. Consequently, the IRI at all near infrared wavelengths were overestimated and the SSA was underestimated (Table 4). Figure 5 plots the ratio of the diffuse radiance to that of P1 at 340 and 940 nm. The strong diffuse radiances at the wavelengths smaller than 870 nm for the PPL geometry in Fig. 4 were due to the large diffuse radiance at the scattering angles more than 90˚. This increase of the diffuse radiances at backward angles became small at the longer

wavelengths and was negligible at 1020 and 2200 nm. The increase of the backward scattering of P2 and P3 for the PPL geometry affects the balance of the forward and backward scattering and leads to the decrease of the ASM and the increase of the RRI, because, as predicted by the Mie theory, the ASM decreases with an increase of the RRI (Hansen and Travis, 1974). On the other hand, the ratio of the diffuse radiances at 940 nm was large at scattering angles smaller than 90˚ and was small at





scattering angles larger than 90°. This was also seen at 2200 nm. This feature and the smoothness constraint for the RRI caused

the overestimation of the ASM and underestimation of the RRI at the near infrared wavelengths (Fig. 3 and Table 4).

The above experiment suggests that the influence of the aerosol vertical profile cannot be ignored in the retrieval using the data for the PPL geometry. In practice, the aerosol vertical profile has large variability. The synergistic approach with lidar is a reasonable solution for this problem (e.g., Kudo et al., 2016).

## 5 Application to real measurements

### 5.1 Application to the measurements at Tsukuba

The sky radiometer data collected from February to October 2018 in Tsukuba were processed by MRI v2 and Skyrad v4.2 and v5 using the following five configurations:

1) ALM-SW: Aerosol physical and optical properties were retrieved from the measurements in the ALM geometry. To assess the sensitivity of the retrievals to the available measurements, a subset of the measuring wavelengths (i.e., only 340, 380, 400,

500, 675, 870, and 1020 nm) was chosen. The PWV and TO3 were not retrieved but were given from the measurements of the radiosonde at 00 UTC and the daily mean of the Brewer spectrophotometer observation.

2) PPL-SW: measurements in the PPL geometry with the same wavelength subset as in ALM-SW.

3) ALM-LW: Aerosol physical and optical properties, PWV, and TO3 were retrieved from the measurements in the ALM geometry. The complete set of data at all the wavelengths from 315 to 2200 nm were used.

4) PPL-LW: measurements in the PPL geometry using the full range of wavelengths.

5) V42: the same dataset as in ALM-SW was employed, but the retrieval method was Skyrad v4.2.

6) V5: the same dataset as in ALM-SW was employed, but the retrieval method was Skyrad v5.

The aerosol physical and optical properties, PWV, and TO3 of the five patterns were compared. In addition, the surface solar irradiance simulated using the retrieved quantities from MRI v2 were compared with the measurements.

### 5.1.1 Aerosols

Figure 6 illustrates the daily means of AOD, Ångström exponent, RRI, IRI, SSA, ASM, LIR, and DEP. The wavelength is 500 nm. The AOD is calculated from the retrieved VSD, RRI, and IRI, therefore it can be considered as a metrics of the degree of "internal closure" of the model itself. The Ångström exponent is calculated from a logarithmic fit from the AOD at the wavelengths of 340, 380, 400, 500, 675, 870, 1020 nm. The AOD was large in spring, from March to May, and the Ångström

exponent was low. This can be explained by transport of dust from China and Mongolia (e.g., Kudo et al., 2010; Kudo et al., 2018). The AOD and Ångström exponent with all configurations agreed well.

The RRI at 500 nm for all the retrievals was large in spring, ranging from 1.45 to 1.55. According to the database of the Hess et al. (1999) and Aoki et al. (2005), the RRI of the dust is about 1.53 at visible wavelengths and larger than that of other aerosols except for the black carbon. The RRI of ALM-SW and PPL-SW were consistent with those of the V42 and V5



through the observation period. However, the RRI of ALM-LW and PPL-LW were lower in February and larger from June to October. The AOD at 500 nm was generally smaller than 0.2 during these periods, and the differences of the RRI are within the retrieval uncertainties of the RRI (Table 3). As mentioned in Sect. 4.1, the refractive index is less sensitive to the diffuse radiances for the low AOD. In particular, the AOD at 1627 and 2200 nm was less than 0.1 in February and June to October. It is difficult to retrieve the refractive index at 1627 and 2200 nm in this condition. The retrieval errors at 1627 and 2200 nm

might affect the RRI at the short wavelengths through the smoothness constraint for the spectral dependency of the RRI.

       The IRI at 500 nm of MRI v2 (ALM-SW, PPL-SW, ALM-LW, and PPL-LW) was larger than those of V42 and V5 in the whole period. Therefore, the SSA of MRI v2 were also smaller than V42 and V5 in the whole period. The previous studies show that Skyrad v4.2 tends to underestimate the IRI and overestimate the SSA, compared with AERONET (Che et al., 2008; Estellés et al., 2012; Khatri et al., 2016). Hence, the IRI and SSA of MRI v2 may be closer to AERONET retrievals.

450       Although the ASM of MRI v2 and V5 at 500 nm were slightly larger than V42, the ASM of all the retrieval patterns had similar values from 0.6 to 0.7. These are within the typical values of the tropospheric aerosols (Dubovik et al., 2002).

       The LIR of MRI v2 ranged from 40 to 100 sr and was larger than those of V42 and V5, i.e. 20 to 40 sr. Observations from the HSRL at Tsukuba in two years shows that the LIR at 532 nm is 68±39.8 sr for spherical particles and 48±27.1 sr for non-spherical particles (Tatarov et al., 2006). Tsukuba is a rural city near Tokyo, and the typical LIR observed in the polluted or urban air are from 40 to 80 sr (Burton et al., 2012; Groß et al., 2015). The LIR of MRI v2 lied within the ranges of the HSRL

observation at Tsukuba and typical polluted air, while those of V42 and V5 were too small. The LIR increases as the SSA decreases. In addition, the non-spherical particles are incorporated in MRI v2 but not in the Skyrad v4.2 and v5. Generally, for the same refractive index and particle size, the phase function of the non-spherical particles at the scattering angle 180˚ is smaller than that of spherical particles, and the LIR of non-spherical particles is larger than that of spherical particles (e.g.,

Dubovik et al., 2006; Kudo et al. 2016). Both the SSA and the particle shape contribute to the difference of the LIR between MRI v2, and Skyrad v4.2 and v5.

       The DEP was large, up to 0.2, in spring because of the transported dust. The difference among the ALM-SW, PPL-SW, ALM-LW, and PPL-LW retrievals was small. The V42 and V5 do not consider non-spherical particles and cannot derive the DEP. Two years observations of the Raman lidar at Tsukuba show the DEP of the dust layer to be about 0.2±0.07 (Sakai

et al., 2003). Considering that our retrieval is a columnar property and includes spherical particles at low altitudes in addition to the dust layer, it is natural that our retrieved DEP is slightly smaller than that of the dust layer directly observed by the lidar.

       Figure 7 shows the spectral dependencies of AOD, VSD normalized by the total volume, RRI, IRI, SSA, ASM, LIR, and DEP. They are represented as means over the whole observation period. The spectral dependencies of the AOD of all the retrievals showed good agreement. The difference of the RRI, ASM, and DEP among all the retrievals were also small.

470       The spectral changes in the IRI and SSA of MRI v2 was small. However, the IRI of V42 and V5 drastically decreased from the near ultraviolet wavelength to the visible wavelength. Hence, the SSA was small at the near ultraviolet wavelengths and was large at the visible and near infrared wavelengths. The differences of the SSA at the visible wavelengths between MRI



v2 and Skyrad v4.2 and v5 could come from the smoothness constraint of the spectral dependency for the IRI. It should be considered that the constraint of MRI v2 are stronger than those of the Skyrad v4.2 and v5.

The LIR of V42 and V5 retrievals were smaller than that from MRI v2 at all the wavelengths. This could be due to the differences of the particle shape and SSA. The lidar ratio of the retrieval for the almucantar geometry (ALM-SW and ALM-LW) at the near ultraviolet and visible wavelengths was about 10 sr smaller than that of the retrieval for the principal plane geometry (PPL-SW and PPL-LW). However, this value is within the retrieval uncertainties (Tables 3 and 4).

Figure 7b is the VSD normalized to the total volume. The normalized VSD from MRI v2 had a similar bi-modal

shape. However, the normalized VSD of V42 had a tri-modal shape. The normalized VSD of V5 had a shape like a mixture of MRI v2 and V42. Although there are no desert regions in Tsukuba, the third mode of V42 appeared in the whole observation period, regardless of seasons. It is unrealistic that almost all the data was contaminated by cirrus clouds. The third mode of the Skyrad v4.2 was also found during other comparisons with the AERONET retrievals (Che et al., 2008; Estellés et al., 2012). If the third mode is real, the microplastics might be a cause because our observational site is neighboring the highway. The

wear and tear from tyres are major source of microplastics, and the particle size range is from 1 nm to 100 μm (Kole et al., 2017; Evangeliou et al. 2020). However, we could not find any data and investigations which relate our results to microplastics. If the third mode is not real, we have to consider the measurement errors of the diffuse radiances at scattering angles near the sun. According to Fig. A2, it is considered that the third mode at radius around 10 μm is created by the diffuse radiances at the scattering angles around 3˚ and at the wavelengths of around 1020 nm. Indeed, the third mode does not appear if we do not

use the diffuse radiances at scattering angles less than 4˚ (Fig. 8). We need to investigate the measurement accuracies of the diffuse radiances at scattering angles near the sun. In the MRI v2 retrieval, the dynamic weight factor depending on the AOD (Eq. 5) weakens the contribution of the diffuse radiances at the near infrared wavelengths to the objective function (Eq. 4). This has the effect to suppress the third mode of the VSD.

We further examined the differences of the ALM-SW, V42, and V5 retrievals because their wavelength subset is the

standard in SKYNET. Figure 9 shows the comparison of AOD from the ALM-SW, V42, and V5 retrievals with the AOD obtained from the direct irradiance (DAOD). The DAOD was calculated by the Skyrad v4.2 software. The AOD of ALM-SW, V42, and V5 are calculated from the retrieved VSD, RRI, and IRI. The AOD of V42 and V5 agreed with the DAOD but the AOD of ALM-SW was slightly larger than the DAOD. The mean differences of the AOD of ALM-SW from the DAOD were less than about 0.02 for all the wavelengths (Table 5). The large AOD of ALM-SW comes from the reason that MRI v2

optimizes the parameters to the transmittances, but the Skyrad v4.2 and v5 optimizes the parameters to the DAOD. In the optimal estimation, the parameters are preferentially optimized to the measurements, which significantly contribute the objective function (Eq. 4). The contributions of the measurements depend on the value of $W$ in Eq. 4. The value of $W$ for the DAOD is 0.01/DAOD in the Skyrad v5. This means that the algorithm tries to optimize the parameters to the range of DAOD ±0.01. In MRI v2, the value of 2 % is given to the transmittance. The relation of the perturbation between the transmittance

($T$) and AOD ($\tau$) is described as follows:





$$T + \Delta T = \exp\{-m(\tau + \Delta\tau)\}, \tag{15}$$

$$\tau + \Delta\tau = -\frac{1}{m}\ln(T + \Delta T) = -\frac{1}{m}\ln T - \frac{1}{m}\ln\left(1 + \frac{\Delta T}{T}\right) = \tau - \frac{1}{m}\ln\left(1 + \frac{\Delta T}{T}\right), \tag{16}$$

$$\frac{\Delta\tau}{\tau} = -\frac{1}{m\tau}\ln\left(1 + \frac{\Delta T}{T}\right) \cong -\frac{1}{m\tau}\frac{\Delta T}{T}, \tag{17}$$

where $\Delta\tau$ and $\Delta T$ are the perturbations for $\tau$ and $T$, $m$ is optical air mass. For the simplicity, the optical depths for the Rayleigh

scattering and gaseous absorption are ignored. $\Delta T/T$ is 0.02 in MRI v2. In the case that $\tau$ is 0.4 and $m$ is 1.2, $\Delta\tau/\tau$ is 0.042 in MRI v2, and 0.025 in the Skyrad v5. The value of $\Delta\tau/\tau$ of MRI v2 is always larger than that of the Skyrad v5 in the range of $m$ from 1 to 2. Therefore, the fitness of the parameters to the DAOD in MRI v2 is weaker than the Skyrad v5.

Figure 10 is the comparison of SSA between ALM-SW, V42, and V5. The SSA approached to similar values when the AOD at 500 nm is more than 0.3. Furthermore, the coefficients of the determination between the ALM-SW and V5 were

higher than the V42. Overall, these features were also seen in the scatter plots of the RRI, IRI, ASM, and LIR (Figs. S1 to S4). Figure 11 illustrates the normalized VSD of ALM-SW, V42, and V5 for the different values of AOD at 500 nm. The third mode of the V42 retrieval decreases when the AOD is more than 0.3. The third mode of the SKYRAD v4.2 is a common feature in the case of low AODs. The normalized VSD of V5 had a closer shape to that of ALM-SW in the case of the high AOD.

Table 6 is a summary of the differences between the ALM-SW, V42, and V5 retrievals for the different grades of the AOD at 500 nm. The AOD, RRI, and ASM of ALM-SW, V42, and V5 agreed. Conversely, the differences of IRI, SSA, and LIR between the ALM-SW, V42 and V52 were larger than the expected uncertainty, however the differences decrease in the case that the AOD was more than 0.3. These differences were caused by the smoothness constraint for the refractive index and size distribution, the particle shape, and the assumption of measurement errors in the retrieval algorithm.

**5.1.2 Water vapor and ozone**

Figure 12 shows the PWV of the ALM-LW and PPL-LW retrieval results, and radiosonde observation at 00 UTC (09 Japan Standard Time). Their seasonal changes agreed well (Fig. 12a). This is also visible from the scatter plot between the ALM-LW and PPL-LW (Fig. 12b). The means and standard deviations of the differences between the retrievals and the radiosonde observation were 0.1±2.7 mm for the ALM-LW and 0.1±3.9 mm for the PPL-LW.

Figure 13 shows the comparison of the TO3 between the ALM-LW and PPL-LW retrievals, and Brewer spectrophotometer observation. The seasonal changes were consistent among the different techniques (Fig. 13a), and the coefficient of determination was good, more than 0.7 (Fig. 13b). However, the TO3 of the ALM-LW and PPL-LW were slightly large. The means and standard deviations of the differences between the retrievals and the Brewer spectrophotometer were 24±25 m atm-cm for the ALM-LW and 23±24 m atm-cm for the PPL-LW.

In both PWV and TO3 retrievals, the difference between the ALM-LW and PPL-LW were small. It should be considered that the influence of the aerosol vertical profile was small. The PWV could be stably provided from the sky





radiometer. However, further investigation is necessary for the calibration of the measurements at 315 nm and the retrieval of the TO3.

### 5.1.3 Surface solar radiation

The global, direct, and diffuse components of the surface solar irradiance calculated from the retrieved aerosol optical properties, PWV and TO3 of the ALM-SW, PPL-SW, ALM-LW, and PPL-LW were compared with the measurements at the Aerological Observatory, an observational station of BSRN (Fig. 14). Considering the measurement errors of the surface solar irradiance (Sect. 2.1), all the calculated irradiances from the retrievals showed good agreements with the measurements. The mean differences between the calculated irradiances from the retrievals and the measurements were about 24 Wm$^{-2}$ (3 %) for

the global irradiance, less than 22 Wm$^{-2}$ (3.0 %) for the direct irradiance, and less than 4 Wm$^{-2}$ (2 %) for the diffuse irradiance. The ALM-LW and PPL-LW results for the direct irradiance were obviously better than the ALM-SW and PPL-SW, i.e. the use of all the wavelengths and the retrieval of water vapor and ozone turns out to be effective.

### 5.2 Comparison with the aircraft in-situ measurement of the SAVEX-D

The MRI v2 was applied to POM01 sky radiometer measurements during the SAVEX-D campaign that took place at Praia in

the Cape Verde archipelago in August 2015 (Estellés et al., 2018). The retrieval results were compared with the in-situ measurements from the Facility for Airborne Atmospheric Measurements (FAAM) atmospheric research aircraft (Ryder et al., 2018). Table 7 shows the aerosol optical properties retrieved from the sky radiometer in the two flights performed. The AOD observed by the sky radiometer at 500 nm was about 0.64±0.10 on 16 August and 0.25±0.04 on 25 August. In both days, the Saharan dust originating from Africa was observed at altitudes from 1 km to 6 km by an airborne lidar (Marenco et al., 2018).

Maritime aerosols were also observed in the boundary layer, but the dust particles were dominant (Ryder et al., 2018). Therefore, the Saharan dust is a major contributor of the retrieved columnar aerosol optical properties from the sky radiometer. The RRI, IRI, SSA, and ASM of MRI v2 showed good agreements with those derived from the aircraft in-situ measurements (Table 7). The sky radiometer reports the optical properties at 500 nm, whereas reports those same properties at 550 nm. Given the short wavelength difference, we think that these quantities are comparable, because the spectral changes in these quantities

are small between 500 and 550 nm (e.g., Hess et al. 1998).

Figure 15 shows the comparison of the VSD in the two flights. In addition to MRI v2, the retrievals of the Skyrad v4.2 and v5 were compared with the in-situ measurement. In the result on 16 August, all the retrievals revealed a good performance for the coarse mode (radius > 0.5 μm). The fine mode (radius < 0.2 μm) of Skyrad v5 was overestimated. The fine mode of Skyrad v4.2 was slightly overestimated but agreed to the in-situ measurement within the mean±one standard

deviation. All the retrievals failed to reproduce the peak of the giant particles at the radius around 10 μm, because the sky radiometer data at wavelengths from 443 to 1020 nm (1627 nm and 2200 nm are not available in the POM01 model) and at scattering angles more than 3˚ do not have information on the giant particles at radius around 10 μm (Sect. 3.1 and Fig. A2). In the result on 25 August 2015, all the retrievals underestimated the VSD at radii larger than 1.0 μm, in particular the v4.2.





For the radius < 0.2 µm, all distributions displayed the same behavior as for 16 August. During the 25 August experiment, the sky radiometer, which was deployed at the ground site of Praia, measured the direct and diffuse radiation in the southern direction. However, the flight was conducted close to the island of Sal, which is located northeast of Praia, at Santiago island. This spatial difference may be a cause of the underestimation of the coarse mode. Overall, the MRI v2 showed good performances for both the fine and coarse modes of the VSD.

The LIR and DEP were not obtained from the aircraft in-situ measurements. Therefore, the retrieved LIR and DEP were compared with the lidar measurements of the Saharan dust during AER-D and SAMUM 2 (Table 7). The LIR of MRI v2 was 45±9 sr and 52±10 sr at 500 nm, and 46±10 sr and 55±10 sr at 443 nm, respectively for 16 and 25 August. The LIR derived from the airborne elastic backscatter lidar during AER-D was 54±8 at 355 nm (Marenco et al. 2018). Although the wavelengths are different, the LIR of MRI v2 was close to that of the airborne lidar. It is shown that Saharan dust has similar values of LIR between 355 and 532 nm (e.g., Groß et al. 2015; Shin et al., 2018). The DEP of MRI v2 was 0.25±0.08 and 0.25±0.01 at 500

nm, respectovely. The DEP was not obtained from the airborne lidar during AER-D. Therefore, the DEP of MRI v2 was compared with the ground-based Raman lidar measurements during the second field campaign of the Saharan Mineral Dust Experiment (SAMUM 2) in the literature. Groß et al. (2011) investigated the LIR and DEP of the Saharan dust over Praia during SAMUM 2. The Raman lidar observations from 28 to 30 January 2008 showed that the LIR was 63±6 sr at both 355 and 532 nm, and the DEP was about 0.3 at 532 nm. The LIR and DEP of MRI v2 were slightly smaller than the Raman lidar

measurements. The MRI v2 retrieval is the columnar property including the maritime aerosols in the boundary layer. Since the LIR of the sea salt is around 20 sr, and the DEP is near zero (e.g., Groß et al., 2015), the sea salt contributes to decrease both the LIR and DEP. Furthermore, Ryder et al. (2018) described that the aspect ratio of the sampled particles in the flights during the AER-D campaign flights was slightly smaller than the typical value of the Saharan dust. These would be the reason that the LIR and DEP of the sky radiometer retrievals were smaller than the lidar observations during SAMUM 2.

In summary, under dusty conditions, the physical and optical properties of the Saharan dust retrieved by MRI v2 were broadly consistent with those of the aircraft in-situ and lidar measurements. For further evaluation, in-situ validations of the different species for other case studies, such as biomass burning aerosols, is necessary.

## 6 Summary and conclusion

We developed a new method, Skyrad MRI v2, to retrieve aerosols columnar properties (volume size distribution, real and
imaginary parts of the refractive index, single-scattering albedo, asymmetry factor, lidar ratio and linear depolarization ratio), water vapor, and ozone from the sky radiometer measurements. The advantages of MRI v2 are a full use of all the sky radiometer wavelengths from 315 to 2200 nm, and the application to two scanning patterns of the measurements in the ALM and PPL geometries. The use of all the wavelengths provides the aerosol properties covering the full short-wave band and enables the more accurate evaluation of the aerosol impacts on the atmospheric radiative budget. The Skyrad v4.2, v5, and
MRI v1 were applicable to only the measurement in the ALM geometry, but we can analyse the measurements in the PPL





geometry by MRI v2. This increases the amount of the observation available particularly at the observation sites at the low latitudes.

We have estimated the retrieval uncertainties of MRI v2 using simulated data with perturbations comparable to the uncertainties of the measured solar direct irradiance, diffuse radiance, and surface albedo. The resulting retrieval errors showed
a dependency on the AOD and were small in the case of AODs larger than 0.2. The absolute values of biases ± uncertainties in the retrieval products at the near ultraviolet, visible, and near infrared wavelengths were less than 0.04 for the AOD, less than 0.05 for the RRI, less than 130 % for the IRI, less than 50 % for the VSD, less than 0.05 for the SSA, less than 0.02 for the ASM, less than 20 for the LIR, and less than 60 % for the DEP. The retrieval errors of the PWV and TO3 were less than 8 mm and 42 m atm-cm, respectively. Furthermore, the influence of the aerosol vertical profile to the retrieval was investigated.
The impact of the aerosol vertical profile on the retrieval from the diffuse radiances in the ALM geometry were small. However, the aerosol vertical profile cannot be ignored in the retrieval from the diffuse radiances in the PPL geometry. The largest impacts of the aerosol vertical profile were observed in the diffuse radiances at the backscattering angles. The increase of the diffuse radiances resulted in the overestimation of the SSA. The increase of the backward scattering relative to the forward scattering resulted in the underestimation of the ASM. The best solution of this problem is the co-located observations of sky
radiometer and lidar, and a synergistic analysis (e.g., Kudo et al. 2016).

The MRI v2 was applied to the measurements at Tsukuba, Japan. The results of aerosol physical and optical properties are compared with the Skyrad v4.2 and v5. The AOD, RRI, ASM showed good agreements. The IRI of MRI v2 was larger than those of the Skyrad v4.2 and v5. Hence, the SSA of MRI v2 was smaller than the Skyrad v4.2 and v5. In previous studies, it has been pointed out that the Skyrad v4.2 tends to overestimate the SSA compared to the AERONET (Che et al., 2008;
Estellés et al., 2012; Khatri et al., 2016). The MRI v2 overcomes this problem. The LIR of MRI v2 was larger than the one from Skyrad v4.2 and was closer to the HSRL measurements. This is because the MRI v2 considers non-spherical particles. The DEP of MRI v2 increased to about 0.2 in the dust cases in spring. The results were close to the Raman lidar measurements of the dust layers. The VSD of MRI v2 had a typical bi-modal shape but the Skyrad v4.2 had a tri-modal shape. This behaviour in Skyrad v4.2 is also pointed out by the works of Che et al. (2008) and Estellés et al. (2012). There is a possibility that the tri-
modal shape was due to the too large diffuse radiances at the scattering angles around 3˚. The problem also was improved in the MRI v2 retrieval. The PWV of MRI v2 agreed with the radiosonde measurements. The TO3 of MRI v2 was larger, about 25 m atm-cm, than the Brewer spectrophotometer measurements, but the seasonal change was consistent. The overestimation would be due to the calibration constant at 315 nm. Furthermore, we investigated the radiative closure of the surface solar irradiance between the measurements of BSRN and simulations using the retrieved aerosol optical properties, PWV, and TO3.
The simulated surface solar irradiances agreed to the measurement, and the mean errors were about +24 Wm$^{-2}$ (+3 %).

The MRI v2 were compared with the data in two flight periods during the SAVEX-D 2015. The aerosol physical and optical properties retrieved in two events of the transported Saharan dust were compared with those derived from the aircraft in-situ measurements of the Saharan dust (Ryder et al., 2018; Marenco et al., 2018). The RRI, IRI, SSA, ASM, VSD, and LIR of MRI v2 agreed well with the aircraft in-situ measurements. Furthermore, the LIR and DEP of MRI v2 were consistent with



those of the Saharan dust observed by the Raman lidar (Groß et al., 2011). The MRI v2 showed good performances for dust. Recently, it is shown that dust is substantially coarser than represented in current global climate models (Kok et al., 2017; Adebiyi and Kok, 2020). Theoretically, we showed that the size distribution at radius greater than 10 μm can be retrieved by using the sky radiometer measurements at 1627 and 2200 nm. Since the MRI v2 can utilize the measurements at 1627 and 2200 nm but the Skyrad v4.2, v5, and MRI v1 cannot, the remote sensing of SKYNET with MRI v2 would be useful to

investigate the dust particles.

MRI v2 brought out the performance of the sky radiometer more than ever, and the radiative closure experiment showed the consistency between the surface solar radiation and the retrieved aerosol, water vapor, and ozone. MRI v2 would be useful for monitoring aerosols, water vapor, and ozone, evaluating their impacts on the surface solar radiation, and validating the satellite products and the aerosol prediction models. For the further development, the co-located observation with lidar is

useful. The aerosol vertical profile is assumed in MRI v2, but the lidar improves it. The depolarization ratio from the lidar and the refractive index and size distribution from the sky radiometer would be useful to improve the scattering model of non-spherical particles, such as dust, soot, and volcanic ash particles. MRI v2 uses the same randomly oriented spheroid model with a fixed aspect ratio distribution as the AERONET, but it is shown that the inconsistencies in the spectral variabilities of the lidar ratio and depolarization ratio of dust particles between the lidar and AERONET sun photometer data, in particular, in

the near ultraviolet wavelength range (Müller et al., 2010). Recently, the various realistic particle models are developed for dust (e.g., Ishimoto et al., 2010; Gasteiger et al., 2011), soot (e.g., Kahnert et al., 2013; Ishimoto et al., 2019), and volcanic ash (e.g., Lindqvist et al., 2011). The application of these models to the sky radiometer and lidar data would improve the remote sensing of aerosols. The sky radiometer covers the major three wavelengths of the lidar, 355, 532, and 1064 nm. This may be an advantage, compared to the AERONET sun photometer. Moreover, the introducing the polarization measurements to the

sky radiometer may help to retrieve the non-spherical particles. MRI v2 is useful to this future study because the vector radiative transfer model is already incorporated into MRI v2.

**Appendices**

**Appendix A. Calibration of 315 nm**

The sky radiometer data at 315 nm was calibrated with reference to the daily mean of TO3 observed by the Brewer

spectrophotometer at the Aerological Observatory. The calibration procedure is:

(1) The AOD at wavelengths other than 315 nm are calculated from the solar direct irradiances.

(2) The AOD at 315 nm is calculated by the extrapolation in the log-log plane of wavelength and AOD.

(3) Select a day in the condition that the daily change in the AOD is small. This is a condition required in the Langley method.

(4) An initial value of the calibration constant at 315 nm is estimated by the Langley method.

(5) Transmittance at 315 nm is calculated using the calibration constant at 315 nm.

(6) The TO3 is estimated by minimizing the following objective function:

$$f(\text{TO3}) = \left(T^{obs} - T(\text{TO3})\right)^{T}\left(T^{obs} - T(\text{TO3})\right), \tag{A1}$$

where $T^{obs}$ is the observed transmittance of (5), and $T(\text{TO3})$ is the calculated transmittance by the same forward model described in Sect. 2.3. That is, the response function of the interference filter, the vertical profiles of pressure, temperature, and ozone concentration are taken into account. This non-linear least square problem is solved by the Gauss-Newton method.

(7) The daily mean of TO3 is calculated.

(8) The daily mean TO3 is compared with that observed by the Brewer spectrophotometer. If the difference is large, the calibration constant is changed manually, and go back to (5). If the difference is small, the process is stopped. The threshold for the difference is defined as 3 m atm-cm, which is about 1 % of the climate value of the TO3, 310 m atm-cm at Tsukuba.

This procedure is not automated now. We subjectively selected 15 days during the observation period at Tsukuba and estimated the calibration constants at 315 nm. Figure A1 shows the result of the calibration constant at 315 nm and the fitting line by the polynomial regression. The calibration constant drastically changed in a year. This feature is also pointed out by Khatri et al. (2014). We were not able to find the cause of this change in this study. Further investigation is necessary. The calibration constant obtained by the polynomial approximation was used in the analysis of Sect. 5.1.

**Appendix B. Relation of the size distribution to the sky radiometer measurements**

Tonna et al. (1995) investigated the relation of the size distribution to the sun/sky photometer measurements, based on the Mie theory. The kernel functions for scattering and extinction relating to the phase function and optical depth were described as follows:

$$K(x) = \frac{3}{8\pi} \frac{[i_1(\theta,x,\widetilde{m}) + i_2(\theta,x,\widetilde{m})]}{x^3}, \tag{A2}$$

$$K_e(x) = \frac{3}{4x} Q_e(x,\widetilde{m}), \tag{A3}$$

where $x$ is size parameter, $i_1$ and $i_2$ are the Mie intensity functions, $Q_e$ is the extinction coefficient, $\theta$ is scattering angle, and $\widetilde{m}$ is the complex refractive index. These functions are normalized to their integral over x and are investigated in Tonna et al. (1995). We also used the same functions but plotted the kernel functions as a function of the particle radius, taking the wavelength and the scattering angle as parameters (Fig. A1). Although Tonna et al. (1995) plotted the kernel functions at the scattering angles of 3, 10, 30, 60, and 120˚, we added the kernel functions at the scattering angle of 2˚ because the sky radiometer measures the diffuse radiances at the scattering angle of 2˚. From the behavior of the kernel functions in Fig. A1, we can see that the sky radiometer model POM-01 measurements, of which wavelength is from 315 to 1020 nm, have the information of the size distribution at the radius from 0.02 to 20 μm. Since the model POM-02 have the measurements at the wavelengths of 1627 and 2200 nm, the radius range is broader, from 0.02 to 30 μm.



**Code availability**

The Skyrad pack MRI version 2 software is available on request by contacting the first author of the paper. The software program is coded in Fortran and is compiled on both Intel and GNU Fortran platforms.

**Data availability**

The sky radiometer data is available on request by contacting the first author of the paper.

**Author contribution**

RK developed the Skyrad MRI v2 code and performed the sky radiometer retrieval experiments. MC and HD intensively tested the Skyrad MRI v2 code on the SKYNET/Europe data. HD, and MM improved the Skyrad MRI v2 code. RK, AU, AY, RN, NO, and HI carried out the observation at Tsukuba. VE, MC, FM, and CR proposed the SAVEX-D experiment. VE, FM, and CR worked in the AER-D and SAVEX-D mission science team implementing the airborne sampling strategy. OI and KN worked in the BSRN observation at Tsukuba, Japan. RK prepared the manuscript with contributions from all co-authors.

**Competing interests**

The authors declare that they have no conflict on interests.

**Acknowledgements**

The authors are grateful to the OpenCLASTR project for allowing us to use the SKYRAD.PACK, RSTAR, and PSTAR in this research. The SAVEX-D campaign airborne data was obtained using the BAe-146-301 Atmospheric Research Aircraft operated by Airtask Ltd and managed by the Facility for Airborne Atmospheric Measurements (FAAM). It was a success thanks to many staff at the Met Office, the University of Leeds, University of Manchester, University of Hertsfordshire, FAAM, Directflight Ltd, Avalon Engineering and BAE Systems.

**Financial support**

Japan Society for the Promotion of Science KAKENHI grant no. 15H01728. EUFAR TNA project (European Union Seventh Framework Programme, grant agreement no. 312609). Spanish Ministry of Economy and Competitiveness and European Regional Development Fund through projects CGL2017-86966-R and RTI2018-096548-B-I00.



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



**Table 1: Configuration of the simulation in the assessment of retrieval uncertainties**

| Aerosol | Water-soluble | Dust | Biomass burning |
|---|---|---|---|
| Radius (µm)/width of fine and coarse modes | 0.118/0.6, 1.17/0.6 | 0.1/0.6, 3.4/0.8 | 0.132/0.4, 4.5/0.6 |
| Volume ratio of fine mode to coarse mode | 2 | 0.066 | 4 |
| Real and imaginary parts of refractive index at all the wavelengths | $1.45-0.0035i$ | $1.53-0.008i$ | $1.52-0.01i$ |
| Single-scattering albedo at 340/500/1020 nm | 0.97/0.97/0.96 | 0.83/0.83/0.87 | 0.88/0.86/0.73 |
| Asymmetry factor at 340/500/1020 nm | 0.68/0.64/0.63 | 0.75/0.75/0.75 | 0.69/0.60/0.40 |
| Lidar ratio at 340/500/1020 nm | 62/53/45 | 78/69/57 | 95/75/31 |
| Linear depolarization ratio at 340/500/1020 nm | 0.03/0.05/0.09 | 0.09/0.15/0.25 | 0.00/0.00/0.00 |
| Aerosol optical depth at 500 nm | Random in the range from 0.0 to 1.0 | | |
| Vertical profile | Constant from 0 to 2 km | | |
| Surface albedo | 0.1 for near ultraviolet and visible wavelengths | | |
| | 0.2 for near infrared wavelengths | | |
| Solar zenith angle | Random in the range from 10 to 70 degree | | |
| Precipitable water vapor | Random in the range from 0 to 100 mm | | |
| Total ozone | Random in the range from 250 to 550 m atm-cm | | |

\* Size distribution, and refractive index of water-soluble, dust, and biomass burning is cited from Dubovik et al. (2000).

Depolarization ratios of the biomass burning model are not zero but less than 0.001.






**Table 2: Random errors of the simulation in the assessment of retrieval uncertainties**

| | |
|---|---|
| Surface albedo | Normally distributed random deviations with the standard deviation of 0.05. |
| Measurement error | |
| Direct irradiance | Normally distributed random errors with the standard deviation of 2 % |
| Diffuse radiance | Normally distributed random errors with the standard deviation of 5 % |

\* The standard deviations for direct irradiance, and diffuse radiances were determined from the works of Uchiyama et al. (2018a), and Kobayashi et al. (2006), respectively.






**Table 3: Means and standard deviations of the retrieval errors**

| Aerosol model | Water-soluble | | Dust | | Biomass Burning | |
|---|---|---|---|---|---|---|
| AOD at 500 nm | ≤ 0.2 | > 0.2 | ≤ 0.2 | > 0.2 | ≤ 0.2 | > 0.2 |
| Aerosol optical depth | | | | | | |
| Near ultraviolet | 0.00±0.02 | 0.01±0.03 | 0.00±0.02 | 0.00±0.01 | 0.00±0.01 | 0.00±0.02 |
| Visible | 0.01±0.01 | 0.01±0.02 | 0.00±0.01 | 0.00±0.01 | 0.00±0.01 | 0.00±0.01 |
| Near infrared | 0.00±0.00 | 0.01±0.02 | 0.00±0.00 | 0.00±0.01 | 0.01±0.02 | 0.01±0.02 |
| Real part of refractive index | | | | | | |
| Near ultraviolet | 0.04±0.07 | 0.01±0.04 | 0.00±0.04 | 0.00±0.05 | 0.00±0.07 | -0.01±0.04 |
| Visible | 0.04±0.06 | 0.01±0.04 | 0.00±0.04 | -0.01±0.04 | 0.00±0.07 | -0.01±0.03 |
| Near infrared | 0.03±0.05 | 0.01±0.03 | -0.01±0.04 | -0.01±0.03 | 0.00±0.06 | -0.01±0.04 |
| Imaginary part of refractive index (%) | | | | | | |
| Near ultraviolet | 800±2460 | 26±62 | 25±155 | 0±24 | 77±309 | -3±21 |
| Visible | 827±2475 | 46±80 | 24±152 | 1±25 | 102±317 | 3±37 |
| Near infrared | 934±2495 | 179±410 | 21±146 | 3±30 | 255±450 | 59±216 |
| Volume size distribution in the radius range around the fine and coarse mode peaks (%) | | | | | | |
| Fine mode | 4±83 | 0±27 | 22±447 | 6±42 | 9±40 | 4±20 |
| Coarse mode | 2±21 | 2±14 | 2±11 | 3±9 | -5±18 | 2±13 |
| Single-scattering albedo | | | | | | |
| Near ultraviolet | -0.07±0.13 | -0.01±0.02 | 0.00±0.07 | 0.00±0.02 | -0.03±0.11 | 0.00±0.02 |
| Visible | -0.08±0.14 | -0.01±0.02 | 0.00±0.07 | 0.00±0.02 | -0.06±0.13 | -0.01±0.04 |
| Near infrared | -0.12±0.18 | -0.05±0.08 | 0.00±0.06 | 0.00±0.02 | -0.13±0.20 | -0.05±0.11 |
| Asymmetry factor | | | | | | |
| Near ultraviolet | 0.01±0.03 | 0.00±0.01 | 0.02±0.04 | 0.01±0.01 | 0.00±0.02 | 0.00±0.01 |
| Visible | 0.00±0.04 | 0.00±0.01 | 0.01±0.03 | 0.00±0.01 | -0.01±0.03 | 0.00±0.01 |
| Near infrared | -0.01±0.04 | 0.00±0.01 | 0.00±0.03 | 0.00±0.01 | 0.00±0.04 | 0.00±0.02 |
| Lidar ratio (sr) | | | | | | |
| Near ultraviolet | 24±54 | 3±8 | 4±34 | 4±12 | 13±51 | -1±11 |
| Visible | 16±33 | 1±5 | 4±32 | -2±10 | 8±28 | 0±9 |
| Near infrared | 13±33 | 3±12 | 0±23 | -3±9 | 24±42 | 7±24 |
| Linear depolarization ratio (%) | | | | | | |
| Near ultraviolet | -3±86 | -4±47 | 13±50 | 7±31 | 374±530 | 240±473 |
| Visible | -10±75 | -12±42 | -4±37 | -9±27 | 154±310 | 90±221 |
| Near infrared | -26±47 | -27±34 | -18±26 | -20±23 | -13±130 | -24±43 |
| Precipitable water vapor (mm) | | | | | | |
| | -0.8±4.1 | -1.5±5.8 | -1.7±4.9 | -1.0±4.3 | -0.9±5.8 | -1.1±4.5 |
| Total ozone (m atm-cm) | | | | | | |
| | -5.0±19.9 | -7.4±34.6 | 1.1±23.9 | 0.6±24.3 | -2.8±22.2 | -0.8±34.7 |



**Table 4: Means of the retrieval errors for the almucantar/principal plane geometries**

| Aerosol model | Dust | | | Biomass burning | | |
|---|---|---|---|---|---|---|
| Altitude of aerosol layer (km) | 0-2 | 2-4 | 4-6 | 0-2 | 2-4 | 4-6 |
| Aerosol optical depth | | | | | | |
| Near ultraviolet | 0.00/0.00 | 0.00/-0.03 | 0.01/-0.04 | 0.00/0.00 | 0.00/-0.02 | 0.00/-0.02 |
| Visible | 0.00/0.00 | 0.00/-0.01 | 0.01/-0.01 | 0.00/0.00 | 0.00/-0.01 | 0.00/-0.01 |
| Near infrared | 0.00/0.00 | 0.00/0.01 | 0.01/0.01 | 0.00/0.00 | 0.00/0.00 | 0.00/0.01 |
| Real part of refractive index | | | | | | |
| Near ultraviolet | 0.01/0.02 | 0.01/0.00 | 0.00/-0.03 | 0.00/0.00 | 0.00/0.04 | 0.00/0.03 |
| Visible | 0.01/0.01 | 0.01/0.00 | 0.00/-0.02 | 0.00/0.00 | 0.00/0.02 | 0.00/0.02 |
| Near infrared | 0.00/0.00 | 0.00/-0.06 | -0.01/-0.06 | 0.00/0.00 | 0.00/-0.03 | 0.00/-0.04 |
| Imaginary part of refractive index (%) | | | | | | |
| Near ultraviolet | 0/-3 | 10/-61 | -3/-70 | -1/-1 | 5/-41 | 5/-55 |
| Visible | -2/-3 | 3/-39 | 0/-53 | -1/-1 | 2/-21 | 2/-35 |
| Near infrared | -5/-5 | -5/22 | -5/13 | 1/-1 | 1/18 | 0/37 |
| Volume size distribution in the radius range around the fine and coarse mode peaks (%) | | | | | | |
| Fine mode | -4/-5 | -7/-15 | -9/1 | -2/-1 | -1/-39 | -1/-37 |
| Coarse mode | 1/1 | 1/11 | 2/16 | 4/7 | 5/9 | 4/24 |
| Single-scattering albedo | | | | | | |
| Near ultraviolet | 0.00/0.00 | -0.01/0.06 | 0.00/0.07 | 0.00/0.00 | 0.00/0.03 | -0.01/0.05 |
| Visible | 0.00/0.00 | 0.00/0.03 | 0.00/0.05 | 0.00/0.00 | 0.00/0.01 | 0.00/0.03 |
| Near infrared | 0.00/0.00 | 0.00/-0.03 | 0.00/-0.02 | 0.00/0.00 | -0.01/-0.06 | 0.00/-0.09 |
| Asymmetry factor | | | | | | |
| Near ultraviolet | 0.00/0.00 | 0.01/-0.01 | 0.01/-0.02 | 0.00/0.00 | 0.00/-0.06 | 0.00/-0.06 |
| Visible | 0.00/0.00 | 0.00/-0.01 | 0.00/-0.02 | 0.00/0.00 | 0.00/0.00 | 0.00/-0.01 |
| Near infrared | 0.00/0.00 | 0.00/0.02 | 0.00/0.01 | 0.00/0.00 | 0.00/0.04 | 0.00/0.06 |
| Lidar ratio (sr) | | | | | | |
| Near ultraviolet | 4/3 | 8/-41 | 6/-43 | -2/-2 | 1/-43 | 2/-45 |
| Visible | 0/-1 | 2/-37 | 2/-40 | 0/0 | 1/-17 | 1/-21 |
| Near infrared | -2/-2 | -3/-8 | -2/-13 | -2/-3 | -1/10 | -1/16 |
| Linear depolarization ratio (%) | | | | | | |
| Near ultraviolet | 21/24 | 17/-67 | 28/-81 | 98/45 | 90/139 | 96/347 |
| Visible | 5/6 | 3/-78 | 5/-87 | 33/-2 | 30/37 | 36/131 |
| Near infrared | -5/-4 | -8/-80 | -8/-89 | -26/-47 | -29/-22 | -25/-30 |
| Precipitable water vapor (mm) | | | | | | |
| | -0.4/-0.4 | -0.3/2.1 | 4.7/1.3 | -0.5/-0.5 | -0.4/1.8 | -0.5/-10.6 |
| Total ozone (m atm-cm) | | | | | | |
| | 1.2/2.3 | 4.4/42.2 | 16.7/45.6 | -0.7/-1.3 | 0.5/27.2 | 0.3/46.1 |






**Table 5: Means and standard deviations of differences between the aerosol optical depth of ALM-SW, V42, and V5, and the DAOD (aerosol optical depth derived from the direct irradiance)**

| Wavelength (nm) | 340 | 380 | 400 | 500 | 675 | 870 | 1020 |
|---|---|---|---|---|---|---|---|
| ALM-SW | 0.02±0.01 | 0.00±0.01 | 0.01±0.01 | 0.02±0.01 | 0.01±0.01 | 0.01±0.01 | 0.01±0.01 |
| V42 | 0.00±0.01 | 0.01±0.01 | 0.00±0.00 | 0.01±0.01 | 0.00±0.00 | 0.00±0.01 | 0.00±0.01 |
| V5 | 0.00±0.00 | 0.00±0.00 | 0.00±0.00 | 0.00±0.00 | 0.00±0.00 | 0.01±0.00 | 0.01±0.01 |






**Table 6: Means and standard deviations of the differences between the ALM-SW retrievals and the V42 (upper) and V5 (lower) retrievals**

| Wavelength (nm) | 340 | 380 | 400 | 500 | 675 | 870 | 1020 |
|---|---|---|---|---|---|---|---|
| **Aerosol optical depth** | | | | | | | |
| AOD(500) < 0.3 | 0.02±0.01 | 0.01±0.01 | 0.01±0.00 | 0.01±0.00 | 0.01±0.00 | 0.00±0.00 | 0.00±0.01 |
| | 0.02±0.01 | 0.00±0.01 | 0.01±0.01 | 0.02±0.01 | 0.01±0.00 | 0.00±0.00 | 0.00±0.01 |
| AOD(500) ≥ 0.3 | 0.02±0.01 | 0.01±0.00 | 0.01±0.00 | 0.01±0.00 | 0.02±0.01 | 0.01±0.01 | 0.01±0.01 |
| | 0.01±0.01 | 0.00±0.01 | 0.01±0.00 | 0.02±0.00 | 0.02±0.01 | 0.01±0.01 | 0.01±0.01 |
| **Real part of refractive index** | | | | | | | |
| AOD(500) < 0.3 | 0.04±0.05 | 0.02±0.05 | 0.02±0.04 | 0.00±0.04 | 0.02±0.03 | 0.00±0.04 | 0.01±0.04 |
| | 0.03±0.05 | 0.00±0.06 | 0.02±0.04 | 0.03±0.04 | 0.03±0.03 | 0.01±0.04 | -0.01±0.05 |
| AOD(500) ≥ 0.3 | 0.01±0.05 | 0.00±0.04 | 0.00±0.04 | -0.01±0.03 | 0.02±0.04 | 0.01±0.04 | 0.02±0.04 |
| | 0.03±0.05 | 0.01±0.05 | 0.03±0.04 | 0.05±0.04 | 0.05±0.04 | 0.05±0.04 | 0.04±0.04 |
| **Imaginary part of refractive index (Ratios of ALM-SW to V42 or V5)** | | | | | | | |
| AOD(500) < 0.3 | 6.7±15.8 | 6.3±12.7 | 14.2±23.1 | 27.8±26.2 | 15.5±20.6 | 20.6±23.1 | 13.7±14.8 |
| | 1.5±2.6 | 1.4±1.3 | 2.7±3.1 | 27.8±61.6 | 12.0±19.2 | 18.3±48.9 | 10.8±47.8 |
| AOD(500) ≥ 0.3 | 2.5±3.0 | 2.5±3.1 | 4.2±6.5 | 9.2±9.5 | 9.2±11.0 | 12.9±11.6 | 11.7±9.4 |
| | 1.2±0.2 | 1.1±0.2 | 1.6±0.8 | 11.7±14.4 | 11.8±44.9 | 12.1±30.4 | 6.6±41.8 |
| **Single-scattering albedo** | | | | | | | |
| AOD(500) < 0.3 | -0.02±0.05 | 0.01±0.07 | -0.06±0.04 | -0.08±0.04 | -0.05±0.05 | -0.07±0.07 | -0.04±0.08 |
| | 0.00±0.02 | 0.00±0.03 | -0.03±0.02 | -0.08±0.03 | -0.06±0.03 | -0.07±0.04 | -0.02±0.06 |
| AOD(500) ≥ 0.3 | -0.02±0.02 | -0.02±0.02 | -0.03±0.02 | -0.05±0.02 | -0.04±0.03 | -0.06±0.02 | -0.06±0.03 |
| | 0.00±0.01 | 0.01±0.01 | -0.01±0.01 | -0.05±0.01 | -0.03±0.02 | -0.05±0.02 | -0.03±0.02 |
| **Asymmetry factor** | | | | | | | |
| AOD(500) < 0.3 | 0.01±0.03 | 0.02±0.02 | 0.02±0.02 | 0.02±0.02 | 0.00±0.02 | 0.01±0.02 | 0.00±0.03 |
| | 0.00±0.02 | 0.01±0.01 | 0.00±0.01 | 0.00±0.01 | 0.00±0.01 | 0.01±0.02 | 0.02±0.02 |
| AOD(500) ≥ 0.3 | 0.01±0.02 | 0.01±0.01 | 0.01±0.01 | 0.02±0.01 | 0.01±0.02 | 0.02±0.02 | 0.02±0.02 |
| | 0.00±0.01 | 0.01±0.01 | 0.00±0.01 | 0.00±0.01 | -0.01±0.01 | 0.00±0.01 | 0.00±0.02 |
| **Lidar ratio (sr)** | | | | | | | |
| AOD(500) < 0.3 | 16±22 | 13±24 | 29±15 | 34±11 | 22±12 | 23±12 | 18±15 |
| | 8±17 | 14±12 | 18±11 | 30±11 | 19±10 | 20±9 | 18±10 |
| AOD(500) ≥ 0.3 | 22±15 | 23±14 | 27±11 | 30±9 | 23±9 | 25±8 | 23±9 |
| | 11±12 | 14±10 | 16±8 | 23±9 | 16±8 | 17±6 | 13±9 |

AOD(500): Aerosol optical depth at 500 nm of the V42.





**Table 7: Aerosol optical properties of the MRI v2 retrievals using the sky radiometer measurements during the SAVEX-D, the aircraft in-situ measurements of Saharan dust in six flights during AER-D, and Raman lidar measurements of the Saharan dust in three days during SAMUM 2**

| | MRI v2 (500 nm) | | AER-D | SAMUM 2 |
|---|---|---|---|---|
| Optical property | 16 August 2015 | 25 August 2015 | (550 nm) | (532 nm) |
| Aerosol optical depth | 0.64±0.10 | 0.25±0.04 | | |
| Real part of refractive index | 1.49±0.03 | 1.47±0.03 | 1.48[*1] | |
| Imaginary part of refractive index | 0.0012±0.0014 | 0.0015±0.001 | 0.0012−0.0030[*1] | |
| Single-scattering albedo | 0.97±0.03 | 0.96±0.02 | 0.91−0.98[*1] | |
| | | | (mean 0.95) | |
| Asymmetry factor | 0.75±0.02 | 0.74±0.02 | 0.74[*1] | |
| Lidar ratio (sr) | 45±9 | 52±10 | 54±8[*2] (355 nm) | 63±6[*3] |
| | (46±10 at 443 nm) | (55±10 at 443 nm) | | |
| Linear depolarization ratio | 0.25±0.08 | 0.25±0.01 | | 0.29−0.31[*3] |

[*1] Aircraft in-situ measurements over the Cape Verde Islands during AER-D in 2015 (Ryder et al. 2018)

[*2] Elastic backscatter lidar measurements over the Cape Verde Islands during AER-D in 2015 (Marenco et al. 2018)

[*3] Raman lidar measurements over the Cape Verde Islands during SAMUM 2 in 2008 (Groß et al. 2011)





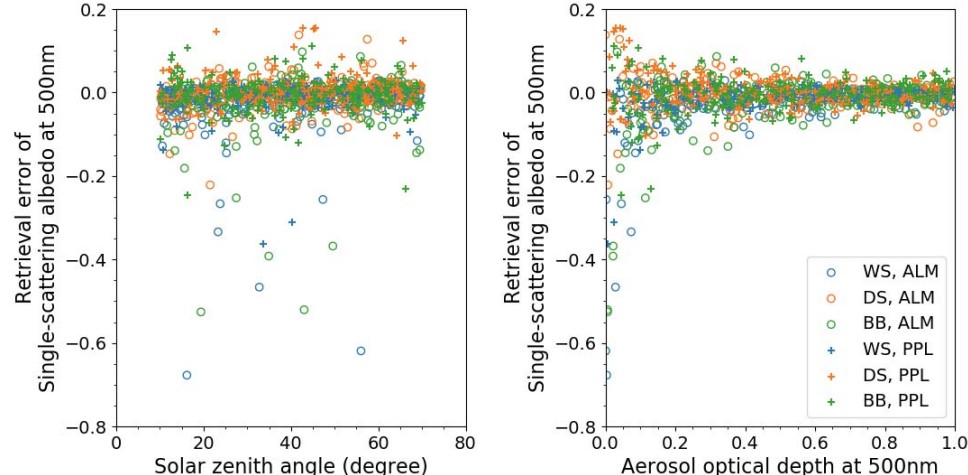

**Figure 1: Dependencies of the retrieval errors of the single-scattering albedo at 500 nm on the solar zenith angle (left) and the aerosol optical depth at 500 nm (right). WS, DS, and BB denote the water-soluble (blue), dust (orange), and biomass burning (green) models, respectively. ALM and PPL denote the scanning patterns of the almucantar (circle) and principal plane (plus) geometries, respectively.**






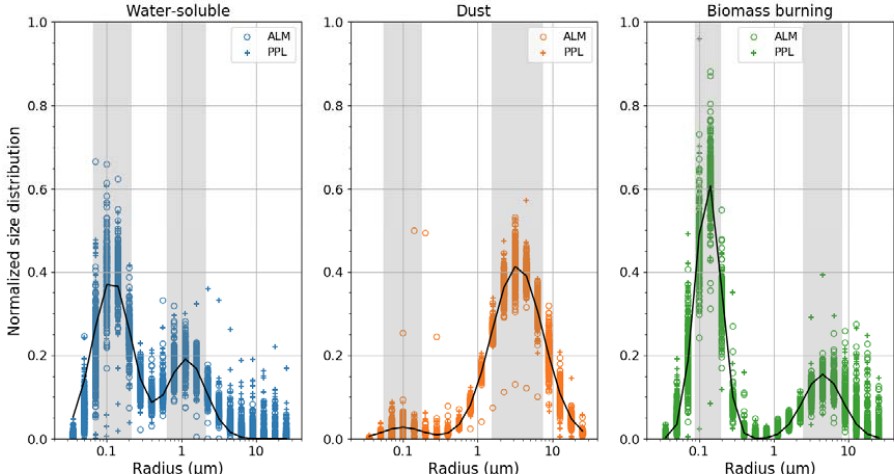

**940** **Figure 2: The retrieved (circle and plus) and simulated (solid line) volume size distributions for water-soluble (left, blue), dust (center, orange), and biomass burning (right, green). The size distribution is normalized to the total volume. Circle and plus symbols indicate that the scanning pattern of simulated data are the almucantar (ALM) and principal plane (PPL) geometries, respectively. The shaded area is the range of the radius around the peak of fine and coarse modes. The range of radius around the peak is defined as mode radius ± one standard deviation (see Table 1).**





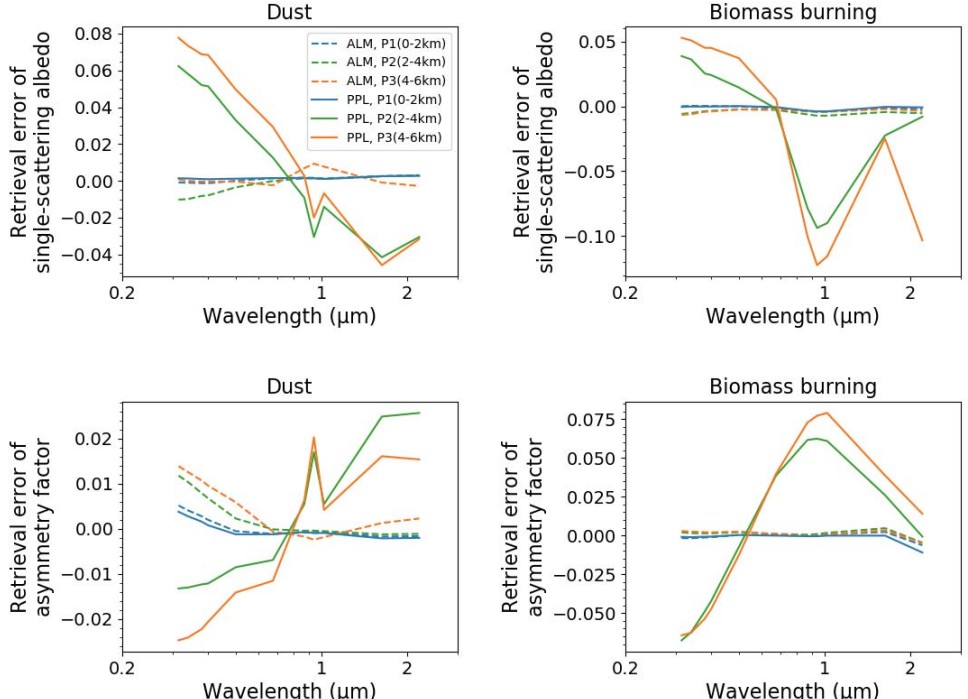


**Figure 3: Retrieval error of the single-scattering albedo (upper) and asymmetry factor (lower) for the simulation data of the dust (left) and biomass burning models (right) with different aerosol vertical profiles of 0 to 2 km (blue), 2 to 4 km (green), and 4 to 6 km (orange). The solid and dashed lines are the retrieval errors for the simulation data in the almucantar (ALM) and principal plane (PPL) geometries, respectively. Note that the y-axis ranges for dust and biomass burning differ in the plots.**






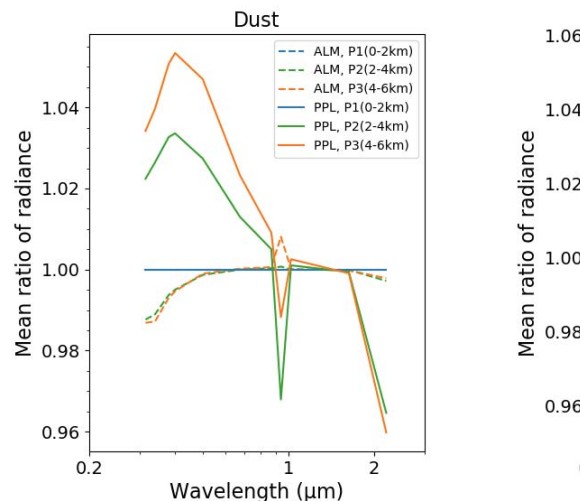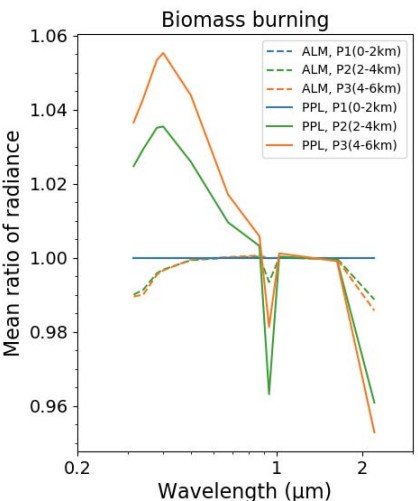

**Figure 4: Mean ratio of the simulated diffuse radiances over the scattering angle with the aerosol vertical profiles of 0 to 2 km (blue), 2 to 4 km (green), and 4 to 6 km (orange) to those with the aerosol vertical profile of 0 to 2 km. Left and right figures are the simulations for the dust and biomass burning models, respectively. The solid and dashed lines are the simulation for the almucantar (ALM) and principal plane (PPL), respectively.**





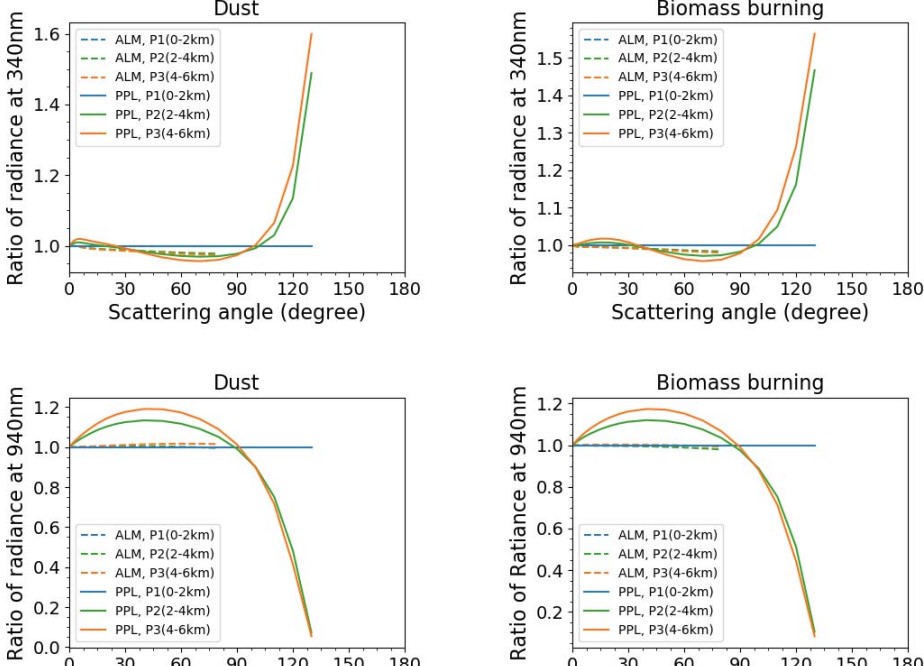


**Figure 5: Ratio of the simulated diffuse radiances at 340 nm (upper) and 940 nm (lower) with different aerosol vertical profiles of 0 to 2 km (blue), 2 to 4 km (green), and 4 to 6 km (orange) to those with an aerosol vertical profile of 0 to 2 km. left and right figures are the simulations for the dust and biomass burning models, respectively. The solid and dashed lines are the simulation for the almucantar (ALM) and principal plane (PPL) geometries, respectively.**




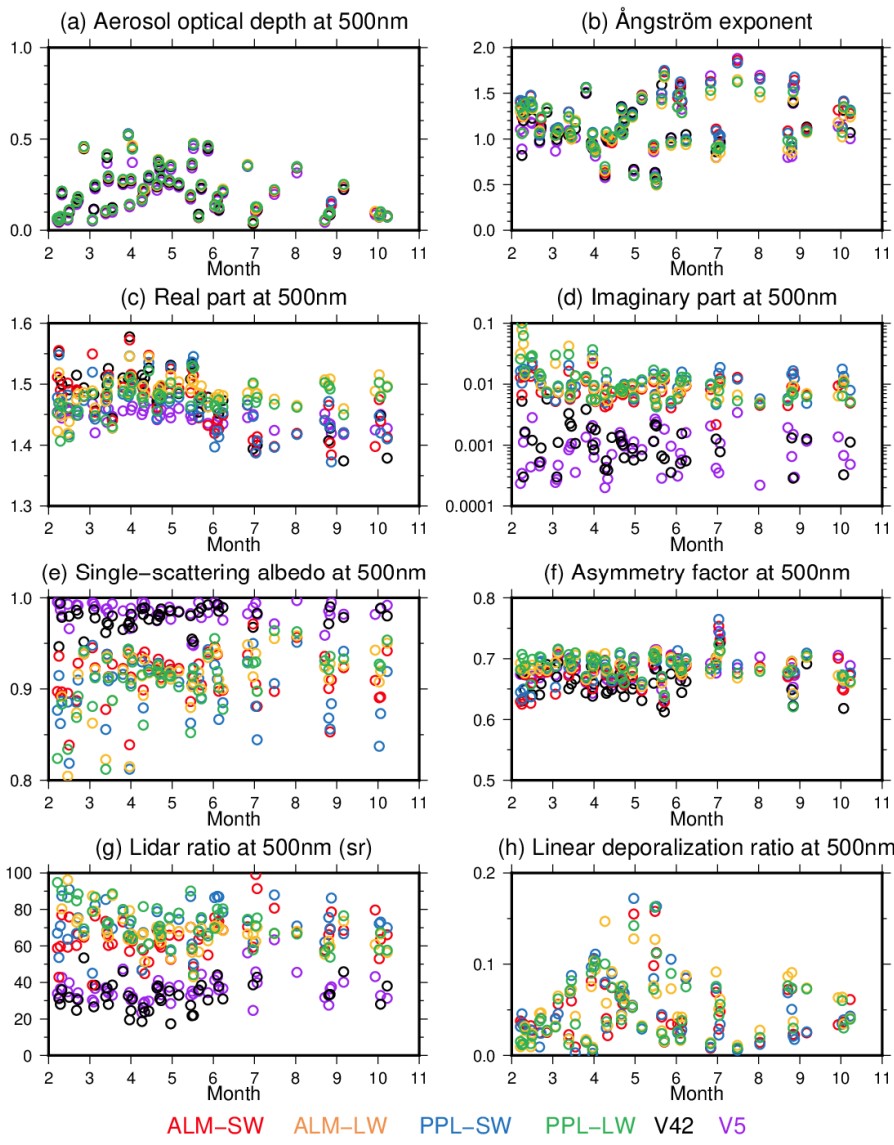

**Figure 6: Daily means of aerosol optical depth (a), Ångström exponent (b), real (c) and imaginary (d) parts of refractive index, single-scattering albedo (e), asymmetry factor (f), lidar ratio (g), and linear depolarization ratio (h). Colors indicate the ALM-SW (red), ALM-LW (yellow), PPL-SW (blue), PPL-LW (green), V42 (black) and V5 (purple) retrieval patterns.**




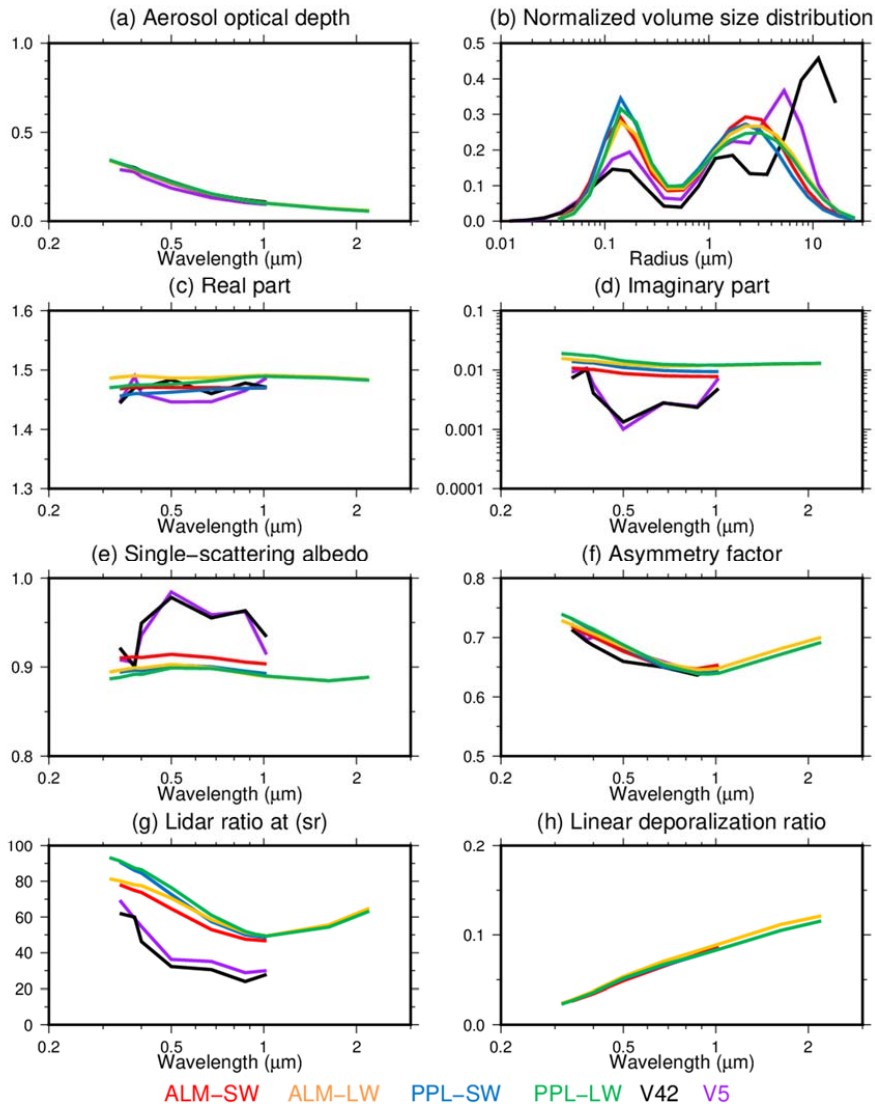

**Figure 7: Means during the whole observation period for aerosol optical depth (a), normalized volume size distribution to total volume (b), real (c) and imaginary (d) parts of refractive index, single-scattering albedo (e), asymmetry factor (f), lidar ratio (g), and linear depolarization ratio (h). Colors indicate the ALM-SW (red), ALM-LW (yellow), PPL-SW (blue), PPL-LW (green), V42 (black) and V5 (purple) retrieval patterns.**



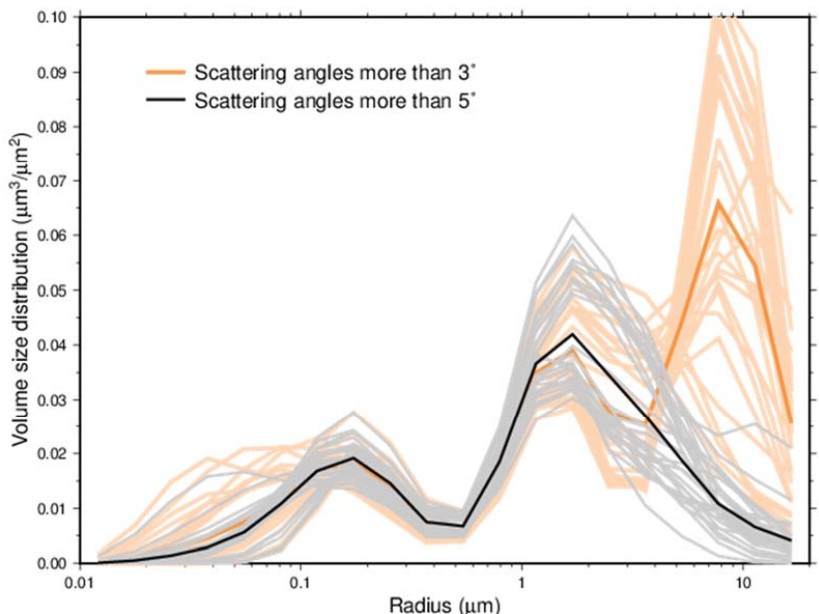

**Figure 8: Volume size distributions retrieved from the diffuse radiances at scattering angles more than 3° (orange) and 5° (gray and black) by the SKYRAD v4.2 in Tsukuba on 14 March 2018. Thick lines are daily means.**



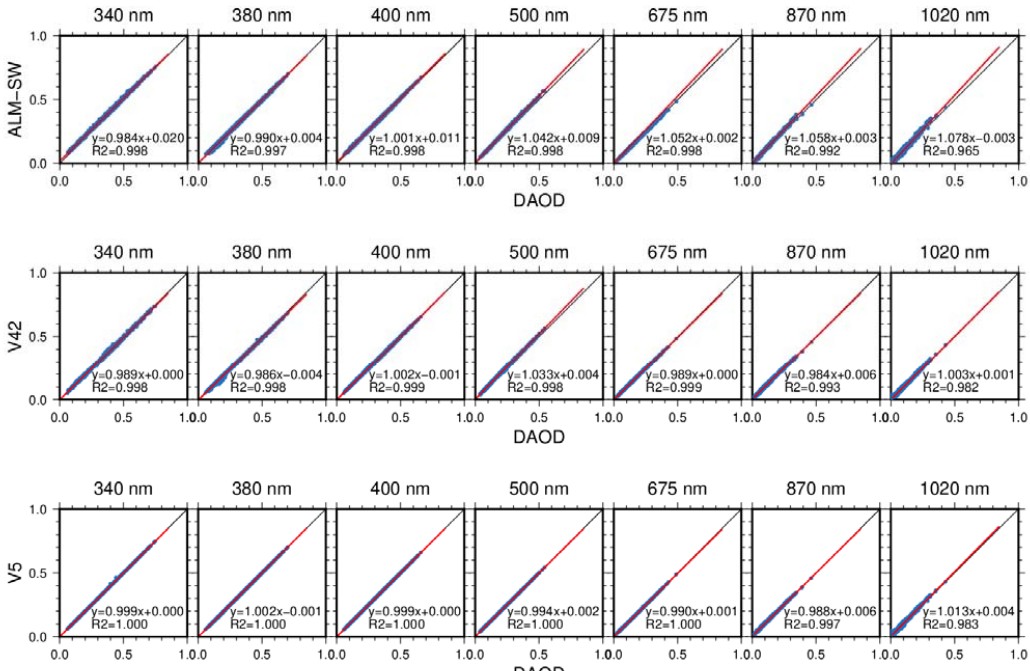

**Figure 9: Comparisons of aerosol optical depth of ALM-SW, V42, and V5 with the aerosol optical depth derived from the direct irradiance (DAOD). The aerosol optical depth of ALM-SW, V42, and V5 are calculated from the retrieved size distribution and refractive index. "y=ax+b" and "R2" are the linear fitting and the coefficient of the determination for all the data.**

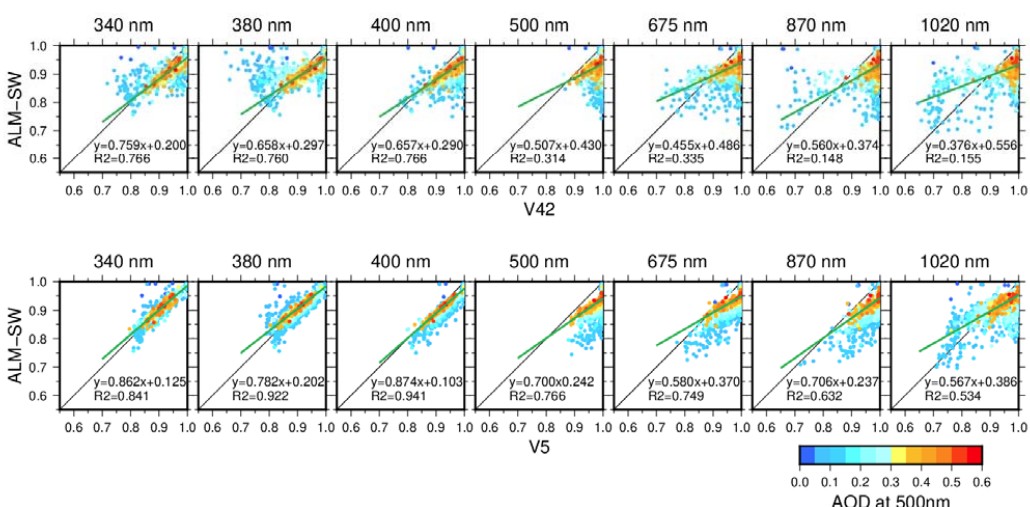

**Figure 10: Comparisons of the single-scattering albedo between ALM-SW, V42, and V5. Colors indicate the aerosol optical depth at 500 nm. "y=ax+b" and "R2" are the linear fitting and the coefficient of the determination for the data of the aerosol optical depth more than 0.3.**





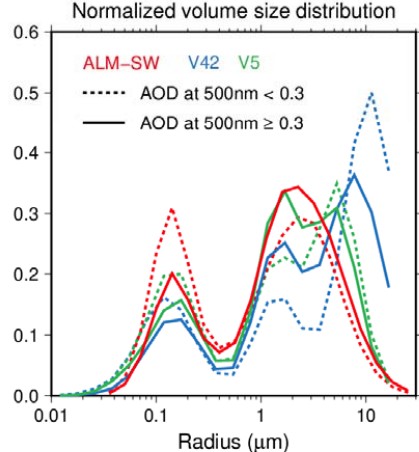


**Figure 11: Means of normalized volume size distributions of the ALM-SW (red), V42 (blue), and V5 (green) for the data of aerosol optical depth at 500 nm more than 0.3 (thick lines) and less than 0.3 (broken lines).**





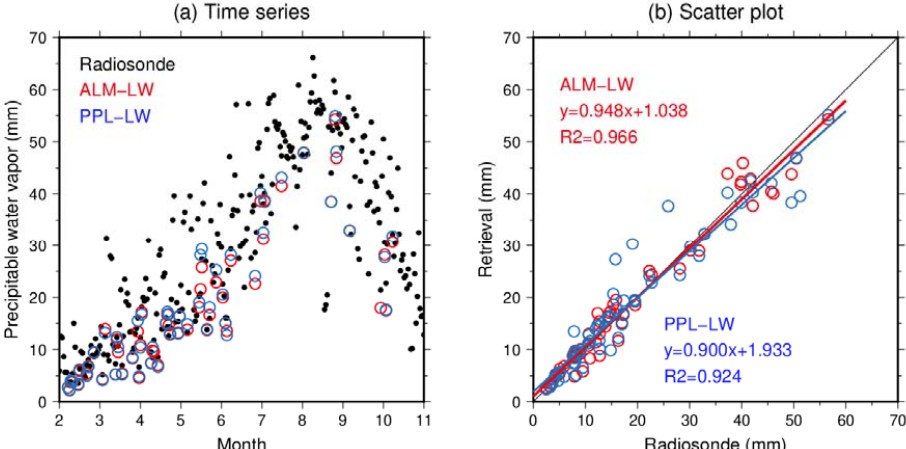

**Figure 12: Seasonal changes (a) and scatter plots (b) of the precipitable water vapor of the ALM-LW (red) and PPL-LW (blue), and**
**radiosonde observation (black). "y=ax+b" and "R2" are the linear fitting and coefficient of determination.**

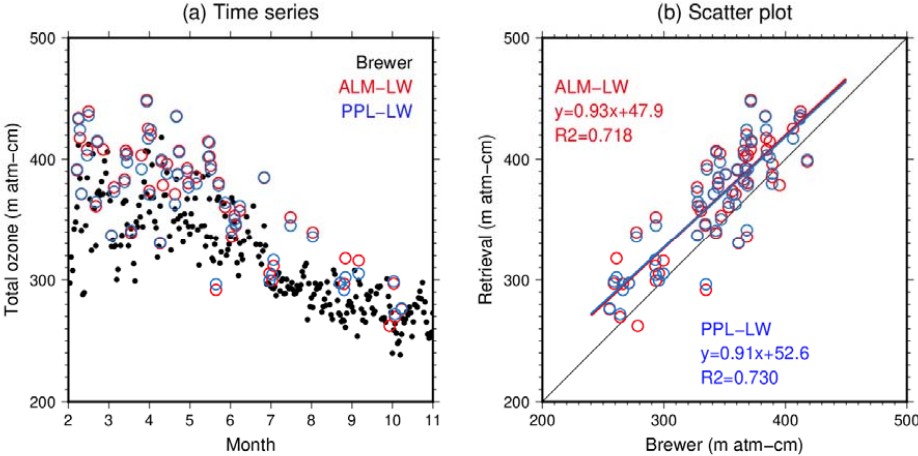

**Figure 13: Seasonal changes (a) and scatter plots (b) of the total ozone of the ALM-LW (red) and PPL-LW (blue), and Brewer**
**spectrophotometer observation (black). "y=ax+b" and "R2" are the linear fitting and coefficient of determination.**



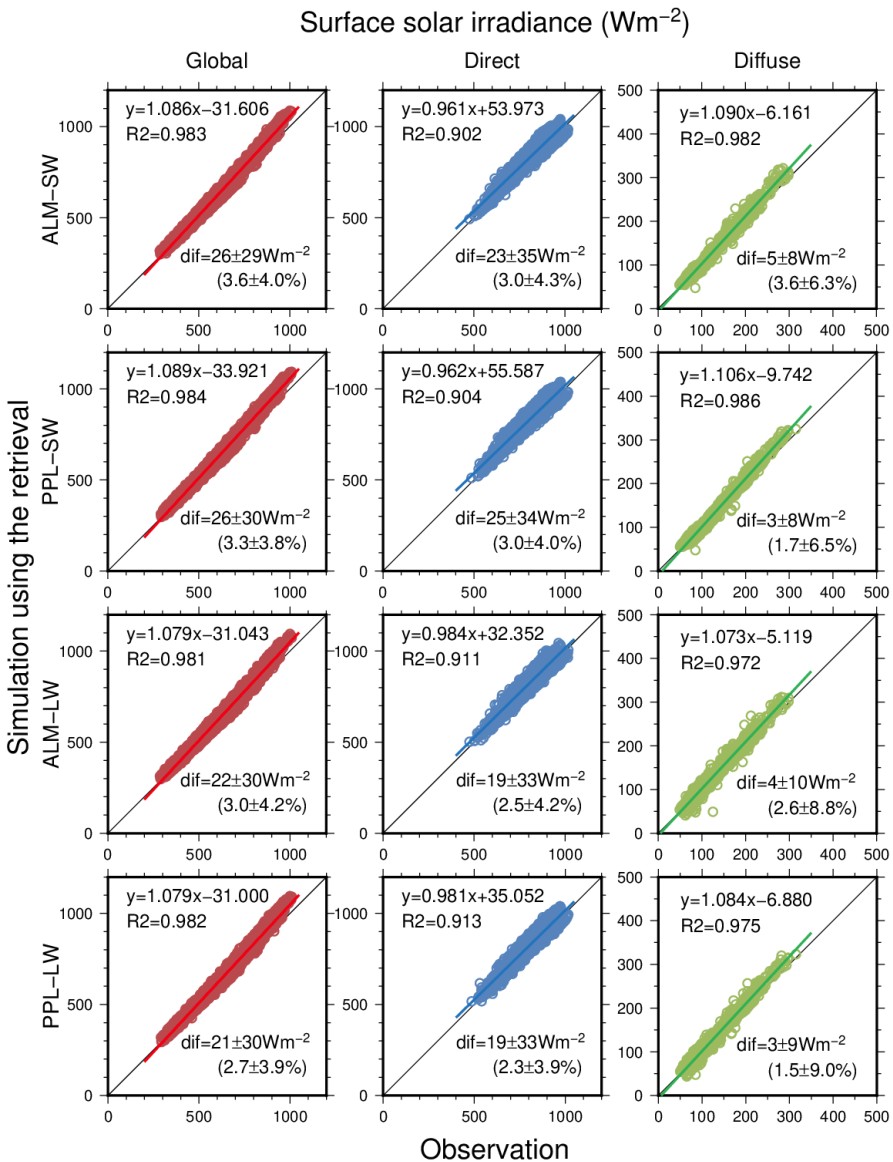

**Figure 14: Comparison of global, direct, and diffuse solar irradiances between the simulations using ALM-SW, ALM-LW, PPL-SW, and PPL-LW retrieval results, and the measurements. "dif" indicates the mean and standard deviation of the differences. "y=ax+b" and "R2" are the linear fitting and coefficient of determination.**

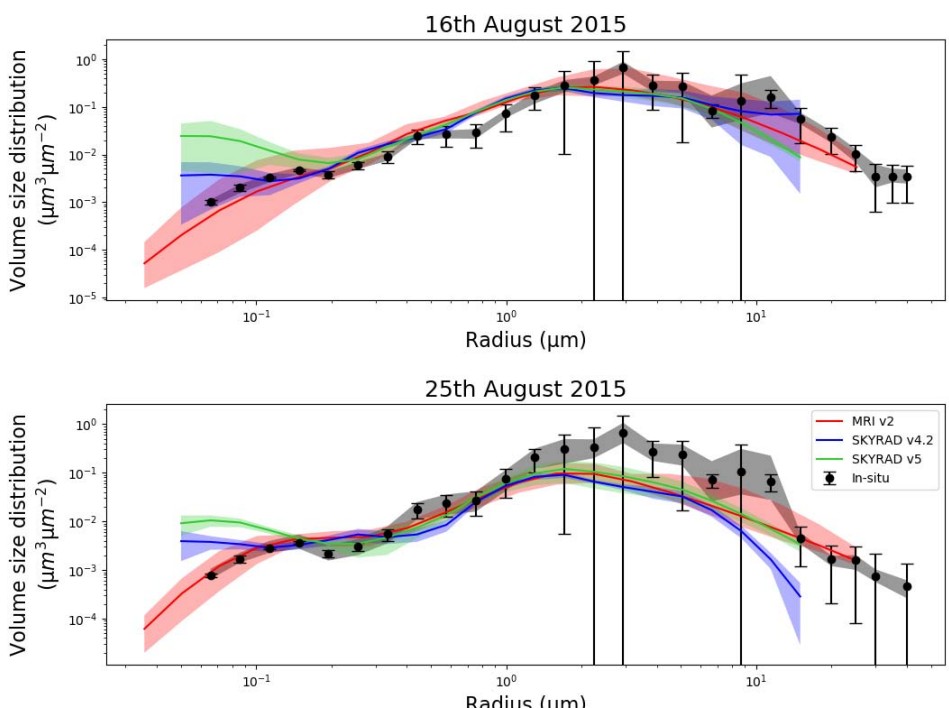

**Figure 15: Volume size distributions of the MRI v2 (red), Skyrad v4.2 (blue), and v5 (green) retrievals and in-situ measurements**
**(black) on 16th (upper) and 25th (lower) August 2015 during the SAVEX-D. Closed circle, error bar, and shaded area of the in-situ**
**measurement indicate mean, measurement uncertainty, and uncertainty due to vertical integration. Solid line and shaded area of**
**MRI v2 indicate the mean, minimum, and maximum. Solid lines and shaded areas of Skyrad v4.2 and v5 indicate the means and**
**standard deviations.**




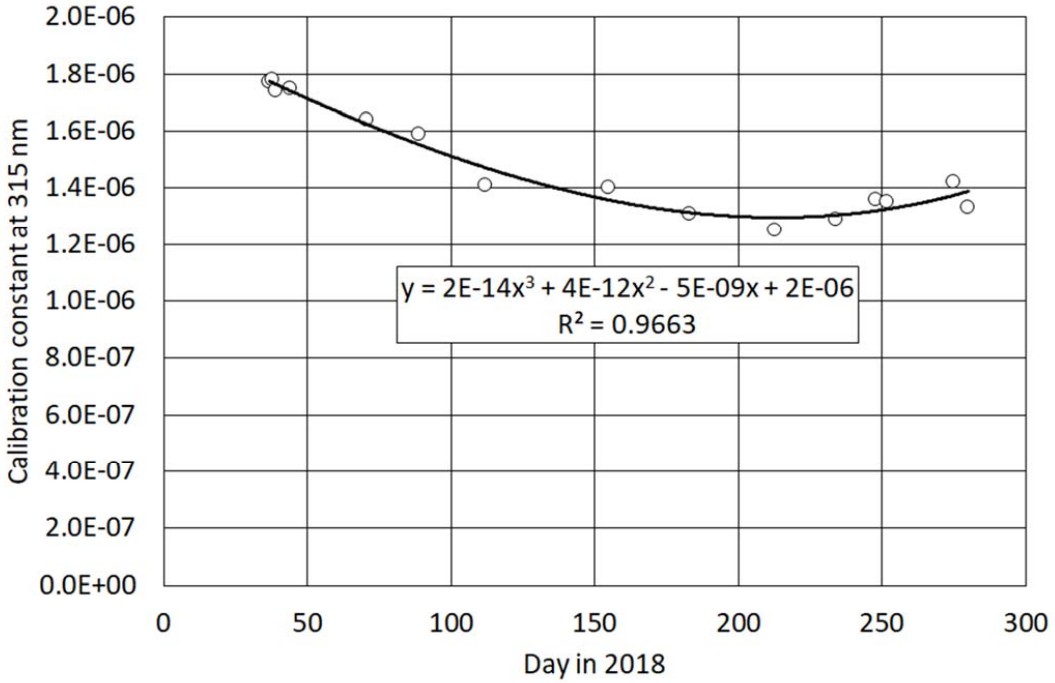

**Figure A1: Estimated calibration constant at 315 nm and the fitting line by the polynomial regression. "R2" is the coefficient of determination.**



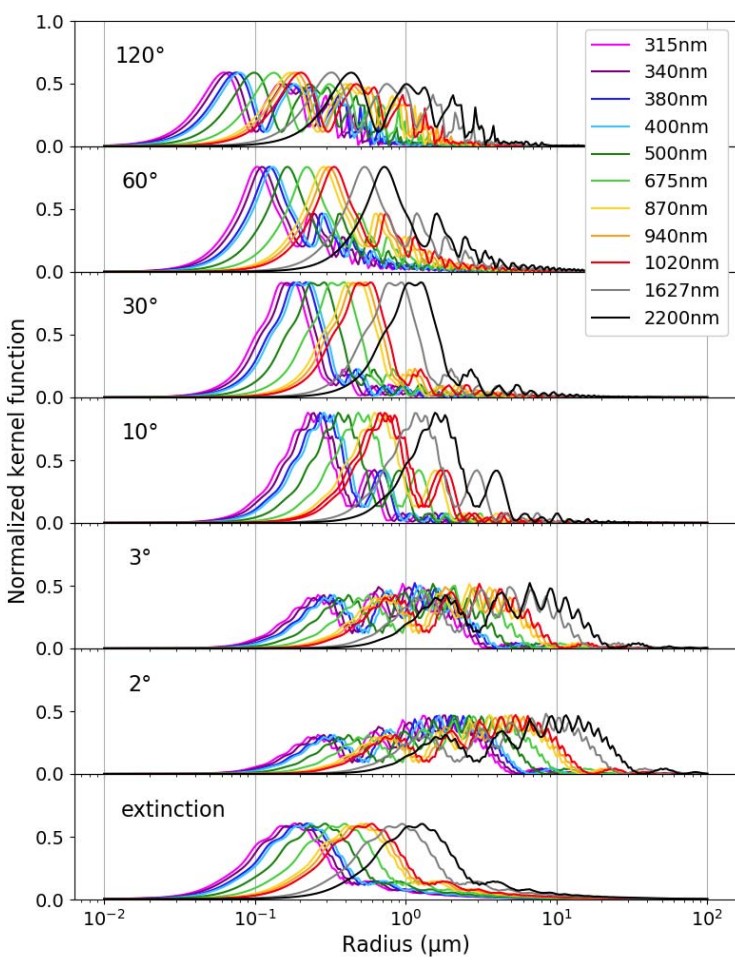

**Figure A2: Normalized kernel functions for extinction and scattering at six scattering angles and at eleven wavelengths. The complex refractive index is assumed to be 1.45-0.0035i.**
