# Peer review of "Optimal use of Prede POM sky radiometer for aerosol, water vapor, and ozone retrievals"

_Atmospheric Measurement Techniques, 2020_

## Editor Comment (EC1) · Stelios Kazadzis (Editor) · 28 Dec 2020

Some comments from the reviewers that came a bit late and you can cosider in the revision phase:

Page 4, line 115- : The authors use the words "irradiance" and "radiance". "Irradiance" normalized by a solid view angle is "radiance". In this paper, diffuse radiance R is defined as a ratio of diffuse irradiance to direct irradiance. However, the definition of radiance is not clear. For example, "the diffuse radiances" in line 122 of this page might be irradiance. I suggest checking these words "irradiance" and "radiance".

Page 5, lines 156-157 : "shading cube" should be "shading ball".

Page 17, eq.(15) : This equation is mathematically incorrect. If they assume this equa-

tion, the relationship between T and AOD is expressed by eq.(17). I understand it, but some descriptions on this matter are kind to readers.

best regards SK

---

## Referee Comment (RC1) · Anonymous Referee #1 · 18 Jan 2021

This manuscript described a new method, Skyrad MRI v2, to retrieve aerosol columnar properties, water vapor, and ozone for the sky radiometer measurements. The new algorithm improved the algorithms of Skyrad v4.2, v5, and MRI v1 in many places, and the results from the new algorithm can bring out the performance of the sky radiometer more than ever. This work is useful for aerosol observation by using sky radiometer. This manuscript can be published after minor revision.

Specific comments: The writing need to be checked. Such as grammar error in Line 35-36.

Line 55, "... aerosol properties, and water vapor and ozone ...", delete "and" before water.

[Figure]

Line 80, "PWV", the full name should be given at the first appearance.

Section 3 Algorithm. In this part, what improvement is made in your inversion algorithm compared to the former algorithms? This may be described clearly, or separately. Is the inversion strategy (section 3.1.1) different from Skyrad v4.2, v5, and MRI v1?

Line 252-253. Did the authors consider the seasonal variation of the vertical profiles of atmosphere parameters?

Section 4.1, Line 339-341. Did this mean that the retrieved data of the new algorithm reduced compared to former algorithms?

Line 372-376. Did the author compare these uncertainties with other algorithms (Such as AERONET)?

Line 491-492. The VSD may be the important improvement of MRI v2 algorithm.

Section 5.1.2. Were these compared to Skyrad V4.2 and V5 algorithms? The aerosol properties retrieved from MRI v2 were improved, then how about the water vapor and ozone?

---

## Referee Comment (RC2) · Anonymous Referee #3 · 21 Jan 2021

This paper is well written and includes new results with a new algorithm for sky radiometer measurements. If the authors consider the following specific comments and revise the draft, the paper will be improved.

(1) Page 5, lines 136-137 : It is better to show the concrete value of view angle, so that discussion on the minimum angle of forward scattering and size distribution will be easily understood. This is related to the comments (2) and (3) below.

(2) Page 8, lines 236-241 : The size limit of large particle might be 20 micron even if the diffuse radiances of 1627 and 2200 nm at scattering angle of 2 degree are used (lines 237-239). What is the reason for the extension of size retrieval range to 30 micron with a scattering angle of 3 degree? Is it appropriate for the size range?

[Figure]

(3) Page 16, 3rd paragraph, Fig.7b and Fig.8 : According to Figure A2, authors discuss the information contained in the wavelength and scattering angle. Why the results from ALM-SW and PPL-SW are not discussed here? The dataset are the same among ALM-SW, PPL-SW, and V42. Why did ALM-SW and PPL-SW succeed and V42 did not for retrieving VSD?

(4) Same part as (3) and Fig. 8 : If the view angle of the sky radiometer is 1.0 or 1.5 degree, forward scattering signal at 2.5 or 2.25 degree is included for the measurement of scattering angle of 3.0 degree. The phase function of forward scattering varies largely and is not linear with scattering angle. The variation of phase function of aerosol at small scattering angle becomes larger with smaller scattering angle, so that the radiance data at 3.0 degree may be overestimated, and then large particles around 10ïĂǎmicron might be overestimated.

(5) Page 17, 5.1.3 Surface solar radiation : The difference between the results with LW and those with SW is quite small even for the direct irradiance as shown in Figure 14. So, the superiority of ALM-LW and PPL-LW as written in the line 546 is not obvious.

(6) Page 18, lines 549 and 566 : POM01 should be POM-01 in line with other pages.

---

## Referee Comment (RC3) · Anonymous Referee #2 · 25 Jan 2021

The work of Kudo et al., is very important in the field of retrieving aerosol and trace gases properties from radiance measurements. A full method consisting of algorithms for these retrievals is presented for the first time and is evaluated using POM data. Results are validated by a closure study and by comparing in situ aircraft data from a campaign. Uncertainties are calculated and causes of errors are treated separately. In general, it is a very good manuscript that will be make an impact on future works. I suggest accepting it for publication after minor revisions.

In particular

A)My main concern is the SSR closure study presented briefly in 5.1.3 and it could be a separate manuscript on its own. I assume it includes only clear sky data (otherwise it would also need cloud properties). I think it would be more useful to present also

some results using V42 V5, in order to convince that the results of the new method are closer to actual conditions. Also, since aerosols (and o3, PWV) accounts for a small part of the variability of Global Irradiance, I suggest presenting not the absolute values, but the differences from measured irradiances. There is a huge potential discussion here, about which how different properties affect the components of SSR, which might be a bit out of the scope of this study. But I think a wider discussion should be reserved for this study, since it is important and complex and not presented in such a brief. B) A general comment is that the discussion on the results and the corresponding figures/tables is very brief. I think an effort should be made to deepen a little the discussion for all figures 6-14 C)Since the use of PP data is a main advantage of the method, I would like to read a small discussion about the suggested measuring schedule of a photometer (regarding PP and ALM , zenith angle and measuring frequency) in respect to the estimated uncertainties, in order to best the exploit method.

L55 in the world⇒around the world L77 please rephrase, for example "low altitude sites" L158 Some details regarding the calibration of pyranometers should be provided. How close to a calibration are the months used in the study? F Section 3.1.2 is mainly discussing and explaining the terms of equation 4. I think it is more reasonable to have the equation and the discussion in the same section. I recommend reorganizing a bit the way the text is separated to sections here. L201 We gave the value→ we assumed or employed the value L209 These unnatural oscillations should be discussed and explained or at least provide some literature examples for them. L339 This means 8.3% of the simulations are rejected. I think these simulations should also be included in the error analysis, since they should reduce the quality of the algorithm. I understand it is difficult to quantify, if there is no output, but It is misleading to ignore such a number of runs, which I understand is expected at this frequency in operational runs. Figure 1 and corresponding discussion. SSA is a parameter that has values up to 1, and variations in the scale of 0.03-0.05 could change the nature (absorbing) of aerosols. Differences up to 0.5-0.6 are unreasonable and should be filtered out by some QC algorithm (these are SSA values of 0.2 or 0.3). Probably you should implement this

plot to a zoomed one in the reasonable range. Still the thickness of the main cloud of data seems to wide to have a robust retrieval, and it doesn't seem to get narrower at higher SZA or AOD (at least from what I can see at this scale). For AOD<0.2 the errors are higher. Differences for shown in table 3 are too high for water soluble aerosols. Probably an acceptable approach could be to ignore SSA retrievals for very low AOD. Finally, I cannot understand why for the lower uncertainties are in the UV region and not in the visible.

L365. After the discussion about the advantage of this method In retrieving bigger particles, I don't understand why this analysis is not up to 30$\mu$m. L390 even for PWV and TO3. I assume this means even for these retrievals. Please restate to be clear. Figure 8 The caption should be rewritten in a clearer way. Figure 10: It seems that MRIv2 retrieves lower SSA even for high AOD, at all wavelengths above 500nm, compared to both V42 and V5. I strongly suggest narrowing the axis range, because all the info is concentrated in the upper right corner and it cannot be seen.

Figure 14: No units for both x and y axis. L 565 giant particles. Since you refer to particles up to 100$\mu$m and even retrieve SD up to 30$\mu$m, It does not add up to name the 10$\mu$m particles as giants. L569-574 It seems that the fact that AOD was 1/3 compared to 16 August could explain a big part of the inconsistencies with the in situ measurements. Figure A2, please express the complex number with same number of significant digits.

Please also note the supplement to this comment:
https://amt.copernicus.org/preprints/amt-2020-486/amt-2020-486-RC3-supplement.pdf
* * *

---

## Author Comment (AC1) · 9 Mar 2021

We thank you for taking the time to review our manuscript and for your useful comments. We revised the manuscript following your comments.

We found the bugs in the simulation of the surface solar irradiance from the retrieved parameters in section 5.1.3. All the surface solar irradiances were calculated again, and the results were improved. Please note that the results and discussions in section 5.1.3 were modified.

Page 4, line 115- : The authors use the words "irradiance" and "radiance". "Irradiance" normalized by a solid view angle is "radiance". In this paper, diffuse radiance R is defined as a ratio of diffuse irradiance to direct irradiance. However, the definition of radiance is not clear. For example, "the diffuse radiances" in line 122 of this page might be irradiance. I suggest checking these words "irradiance" and "radiance".

We checked the words through the manuscript (lines 123, 142), and changed them where necessary.

Page 5, lines 156-157 : "shading cube" should be "shading ball".

We corrected it (line 160).

Page 17, eq.(15) : This equation is mathematically incorrect. If they assume this equation, the relationship between T and AOD is expressed by eq.(17). I understand it, but some descriptions on this matter are kind to readers.

We changed the equations (lines 545-549). In order to discuss the relation of perturbations between the optical depth and transmittance, the differential was used in the revised manuscript. The result is same as previous one.

[revised manuscript text omitted]

(a) Aerosol optical depth
(b) Normalized volume size distribution
(c) Real part
(d) Imaginary part
(e) Single-scattering albedo
(f) Asymmetry factor
(g) Lidar ratio at (sr)
(h) Linear deporalization ratio

ALM–SW  ALM–LW  PPL–SW  PPL–LW  V42  V5

[Figure]

**Figure 7: Means during the whole observation period for aerosol optical depth (a), normalized volume size distribution to total volume (b), real (c) and imaginary (d) parts of refractive index, single-scattering albedo (e), asymmetry factor (f), lidar ratio (g), and**

1070 **linear depolarization ratio (h). Colors indicate the ALM-SW (red), ALM-LW (yellow), PPL-SW (blue), PPL-LW (green), V42 (black) and V5 (purple) retrieval patterns.**

[Figure]

[Figure]

(a) V42                    (b) ALM_SW

1075

**Figure 8:**  Retrieved volume size distribution in Tsukuba on 14 March 2018. (a) The results of the Skyrad v4.2 using the diffuse radiances at scattering angles more than 3° (orange) and 5° (gray and black). (b) The results of MRI v2 using the diffuse radiances at scattering angle more than 3°, and with (gray and black) and without (orange) the dynamic weight factor of Eq. (5). The thick lines are daily means.

[Figure]

1080   **Figure 9: Comparisons of aerosol optical depth of ALM-SW, V42, and V5 with the aerosol optical depth derived from the direct irradiance (DAOD). The aerosol optical depth of ALM-SW, V42, and V5 are calculated from the retrieved size distribution and refractive index. "y=ax+b" and "R2" are the linear fitting and the coefficient of the determination for all the data.**

[Figure]

[Figure]

1085 **Figure 10: Comparisons of the single-scattering albedo between ALM-SW, V42, and V5. Colors indicate the aerosol optical depth at 500 nm. "y=ax+b" and "R2" are the linear fitting and the coefficient of the determination for the data of the aerosol optical depth more than 0.3.**

[Figure]

**Figure 11: Means of normalized volume size distributions of the ALM-SW (red), V42 (blue), and V5 (green) for the data of aerosol**
1090 **optical depth at 500 nm more than 0.3 (thick lines) and less than 0.3 (broken lines).**

[Figure]

**Figure 12: Seasonal changes (a) and scatter plots (b) of the precipitable water vapor of the ALM-LW (red) and PPL-LW (blue), and radiosonde observation (black). "y=ax+b" and "R2" are the linear fitting and coefficient of determination.**

[Figure]

1095    **Figure 13: Seasonal changes (a) and scatter plots (b) of the total ozone of the ALM-LW (red) and PPL-LW (blue), and Brewer spectrophotometer observation (black). "y=ax+b" and "R2" are the linear fitting and coefficient of determination.**

[Figure]

[Figure]

Surface solar irradiance

**Figure 14: Comparison of global, direct, and diffuse solar irradiances between the simulations using ALM-SW, ALM-LW, PPL-SW, PPL-LW, V42, and V5 retrieval results, and the measurements. "dif" indicates the mean and standard deviation of the differences. "y=ax+b" and "R2" are the linear fitting and coefficient of determination.**

[Figure]

**Figure 14: _Differences_ of global _(red)_, direct _(blue)_, and diffuse _(green)_ solar irradiances between the simulations using ALM-SW, ALM-LW, PPL-SW,  PPL-LW, _V42, and V5_ retrieval results, and the measurements, _as a function of the aerosol optical depth at 500 nm_.**

[Figure]

**Figure 15: Volume size distributions of the MRI v2 (red), Skyrad v4.2 (blue), and v5 (green) retrievals and in-situ measurements (black) on 16th (upper) and 25th (lower) August 2015 during the SAVEX-D. Closed circle, error bar, and shaded area of the in-situ measurement indicate mean, measurement uncertainty, and uncertainty due to vertical integration. Solid line and shaded area of MRI v2 indicate the mean, minimum, and maximum. Solid lines and shaded areas of Skyrad v4.2 and v5 indicate the means and standard deviations.**

[Figure]

**Figure A1: Estimated calibration constant at 315 nm and the fitting line by the polynomial regression. "R2" is the coefficient of determination.**

[Figure]

1115

**Figure A2: Normalized kernel functions for extinction and scattering at six scattering angles and at eleven wavelengths. The complex refractive index is assumed to be 1.45-3. 5×10⁻³i.**

---

## Author Comment (AC2) · 9 Mar 2021

Referee #1

This manuscript described a new method, Skyrad MRI v2, to retrieve aerosol columnar properties, water vapor, and ozone for the sky radiometer measurements. The new algorithm improved the algorithms of Skyrad v4.2, v5, and MRI v1 in many places, and the results from the new algorithm can bring out the performance of the sky radiometer more than ever. This work is useful for aerosol observation by using sky radiometer. This manuscript can be published after minor revision.

We thank you for taking the time to review our manuscript and for your useful comments. We revised the manuscript following your comments.

We found the bugs in the simulation of the surface solar irradiance from the retrieved parameters in section 5.1.3. All the surface solar irradiances were calculated again, and the results were improved. Please note that the results and discussions in section 5.1.3 were modified.

Specific comments: The writing need to be checked. Such as grammar error in Line 35-36.

We checked all the text again.

Line 55, ". . . aerosol properties, and water vapor and ozone . . .", delete "and" before water.

We corrected it (line 55).

Line 80, "PWV", the full name should be given at the first appearance.

We added the full name (line 81).

Section 3 Algorithm. In this part, what improvement is made in your inversion algorithm compared to the former algorithms? This may be described clearly, or separately. Is the inversion strategy (section 3.1.1) different from Skyrad v4.2, v5, and MRI v1?

The mathematical equation of the cost function of MRI v2 (Eq. 4) is similar to those of Skyrad v4.2, v5, and MRI v1, but the priori constraints are different. MRI v1, v2 and Skyrad v4.2 employ the smoothness constraints for the real and imaginary parts of the refractive index and the size distribution, whereas Skyrad v5 does not use them. The MRI v1 and Skyrad v5 employ a priori estimates for the refractive index (real and imaginary) and the size distribution. We added these descriptions to the revised manuscript (lines

296-300).

Line 252-253. Did the authors consider the seasonal variation of the vertical profiles of atmosphere parameters?

Yes. The vertical profiles of the pressure and temperature from the radiosonde observation at 00 UTC are used (line 268).

Section 4.1, Line 339-341. Did this mean that the retrieved data of the new algorithm reduced compared to former algorithms?

No. We do not know in how many cases the previous methods fail to retrieve the aerosol parameters from the simulation because we did not apply the previous methods to the simulated data.

Line 372-376. Did the author compare these uncertainties with other algorithms (Such as AERONET)?

Dubovik et al. (2000) showed the uncertainties of the AERONET retrieval. But we cannot simply compare our results with the results of Dubovik et al. (2000). They evaluated the uncertainties by a different approach from ours. They separate the error factors and evaluate the uncertainties for each error factor. Their study is very important as a basic study of the retrieval from the sun-sky photometer. In our study, we assumed realistic situations and evaluated the uncertainties due to the combined influences of the multiple error factors.

Dubovik, O., Smirnov, A., Holben, B. N., King, M. D., Kaufman, Y. J., Eck, T. F., and Slutsker, I.: Accuracy assessments of aerosol optical properties retrieved from aerosol robotic network (AERONET) sun and sky radiance measurements, J. Geophys. Res. Atmos., 105, 9791–9806, https://doi.org/10.1029/2000JD900040, 2000.

Section 5.1.2. Were these compared to Skyrad V4.2 and V5 algorithms? The aerosol properties retrieved from MRI v2 were improved, then how about the water vapor and ozone?

Skyrad V4.2 and V5 cannot retrieve the water vapor and ozone. We added a clear description to the section 5.1.2 (line 574).

[revised manuscript text omitted]

---

## Author Comment (AC3) · 9 Mar 2021

Referee #3

This paper is well written and includes new results with a new algorithm for sky radiometer measurements. If the authors consider the following specific comments and revise the draft, the paper will be improved.

We thank you for taking the time to review our manuscript and for your useful comments. We revised the manuscript following your comments.

We found the bugs in the simulation of the surface solar irradiance from the retrieved parameters in section 5.1.3. All the surface solar irradiances were calculated again, and the results were improved. Please note that the results and discussions in section 5.1.3 were modified.

(1) Page 5, lines 136-137 : It is better to show the concrete value of view angle, so that discussion on the minimum angle of forward scattering and size distribution will be easily understood. This is related to the comments (2) and (3) below.

We added the value of view angle (lines 139-141).

(2) Page 8, lines 236-241 : The size limit of large particle might be 20 micron even if the diffuse radiances of 1627 and 2200 nm at scattering angle of 2 degree are used (lines 237-239). What is the reason for the extension of size retrieval range to 30 micron with a scattering angle of 3 degree? Is it appropriate for the size range?

We added the reason to the revised manuscript (lines 251-255). The size limit is 20 micron if the measurements at scattering angle of 2 degree are not used. We constrain the size distribution at both ends of the radius range to low values by the smoothness constraint of Eq. 10. Therefore, the size distribution at the radius larger than 20 micron is necessary.

(3) Page 16, 3rd paragraph, Fig.7b and Fig.8 : According to Figure A2, authors discuss the information contained in the wavelength and scattering angle. Why the results from ALM-SW and PPL-SW are not discussed here? The dataset are the same among ALM-SW, PPL-SW, and V42. Why did ALM-SW and PPL-SW succeed and V42 did not for retrieving VSD?

We added the results from ALM-SW and described why ALM-SW succeeded in the revised manuscript (lines 525-528, Figure 8).

(4) Same part as (3) and Fig. 8 : If the view angle of the sky radiometer is 1.0 or 1.5 degree, forward scattering signal at 2.5 or 2.25 degree is included for the measurement of

scattering angle of 3.0 degree. The phase function of forward scattering varies largely and is not linear with scattering angle. The variation of phase function of aerosol at small scattering angle becomes larger with smaller scattering angle, so that the radiance data at 3.0 degree may be overestimated, and then large particles around 10ï˜A ˘amicron might be overestimated.

At the beginning of this study, we had the same opinion. Therefore, we developed the forward model including the variation of the radiances within the field of view and used it for the retrieval. However, the retrieved size distribution was not improved, and the third mode appeared. Therefore, we thought that the diffuse radiances measured near the sun have some bias errors.

(5) Page 17, 5.1.3 Surface solar radiation : The difference between the results with LW and those with SW is quite small even for the direct irradiance as shown in Figure 14. So, the superiority of ALM-LW and PPL-LW as written in the line 546 is not obvious.
We changed "obviously" to "slightly" (line 601).

(6) Page 18, lines 549 and 566 : POM01 should be POM-01 in line with other pages.
Thanks, we corrected them (lines 639, 656).

[revised manuscript text omitted]

---

## Author Comment (AC4) · 9 Mar 2021

Referee #2

The work of Kudo et al., is very important in the field of retrieving aerosol and trace gases properties from radiance measurements. A full method consisting of algorithms for these retrievals is presented for the first time and is evaluated using POM data. Results are validated by a closure study and by comparing in situ aircraft data from a campaign. Uncertainties are calculated and causes of errors are treated separately. In general, it is a very good manuscript that will be make an impact on future works. I suggest accepting it for publication after minor revisions.

We thank you for taking the time to review our manuscript and for your useful comments. We revised the manuscript following your comments.

We found the bugs in the simulation of the surface solar irradiance from the retrieved parameters in section 5.1.3. All the surface solar irradiances were calculated again, and the results were improved. Please note that the results and discussions in section 5.1.3 were modified.

In particular

A) My main concern is the SSR closure study presented briefly in 5.1.3 and it could be a separate manuscript on its own. I assume it includes only clear sky data (otherwise it would also need cloud properties). I think it would be more useful to present also some results using V42 V5, in order to convince that the results of the new method are closer to actual conditions. Also, since aerosols (and o3, PWV) accounts for a small part of the variability of Global Irradiance, I suggest presenting not the absolute values, but the differences from measured irradiances. There is a huge potential discussion here, about which how different properties affect the components of SSR, which might be a bit out of the scope of this study. But I think a wider discussion should be reserved for this study, since it is important and complex and not presented in such a brief.

- Only the clear sky data was used. I described this in section 2.1 of the revised manuscript (line 156).
- The results using the V42 and V5 retrievals were added to the revised manuscript (Figure 14, lines 594-606).
- We added Figure 15 which plots the relative differences of the global, direct, and diffuse irradiances as a function of the AOD, and the relation between the aerosol optical properties and the differences of the irradiances were discussed (Figure 15, lines 607-636).
- We agree that there is a huge potential discussion, but further studies are necessary to

understand the inconsistency between the simulated and measured irradiances. The main objective in the present study is to show the performance of new method. Your suggestions are important, and we will continue to investigate on this topic.

B) A general comment is that the discussion on the results and the corresponding figures/tables is very brief. I think an effort should be made to deepen a little the discussion for all figures 6-14

We added the discussion about the SSA (lines 475-479, 556-560), PWV (lines 573-583), and the radiative closure (lines 594-636). We need more specific comments for further improvements.

C) Since the use of PP data is a main advantage of the method, I would like to read a small discussion about the suggested measuring schedule of a photometer (regarding PP and ALM, zenith angle and measuring frequency) in respect to the estimated uncertainties, in order to best the exploit method.

The PP and ALM should be conducted every time. The PP data, in particular 940 nm is affected by the aerosol vertical profile. Conversely, we might be able to estimate the aerosol vertical profile by the synergy of PP and ALM data. We described this in the manuscript (lines 706-711).

L55 in the world => around the world

Thanks, we corrected it (line 55).

L77 please rephrase, for example "low altitude sites"

Thanks, we corrected it (line 77).

L158 Some details regarding the calibration of pyranometers should be provided. How close to a calibration are the months used in the study?

The instruments are regularly calibrated once in 5 years. The pyrheliometer is calibrated in January 2017, and the pyranometer is calibrated in July 2016. These were added to the revised manuscript (line 162).

Section 3.1.2 is mainly discussing and explaining the terms of equation 4. I think it is more reasonable to have the equation and the discussion in the same section. I recommend reorganizing a bit the way the text is separated to sections here.

Since the terms of Eq. (4) are explained in the separate sections, 3.1.2, 3.1.3, and 3.1.4,

we'd like to keep this structure of the text. For the readability, we added the description that the details of each term in Eq. (4) are shown in the sections from 3.1.2 to 3.1.4 (lines 194-198).

L201 We gave the value => we assumed or employed the value
Thanks, we corrected it (line 210).

L209 These unnatural oscillations should be discussed and explained or at least provide some literature examples for them.
There are no literature describing the unnatural oscillations, but there is a literature describing the low signal-to-noise ratio of 1627 and 2200 nm. The unnatural oscillation is due to the low sensitivity of the detector at 1627 and 2200 nm. We added this (line 217).

L339 This means 8.3% of the simulations are rejected. I think these simulations should also be included in the error analysis, since they should reduce the quality of the algorithm. I understand it is difficult to quantify, if there is no output, but It is misleading to ignore such a number of runs, which I understand is expected at this frequency in operational runs.
The quality control is necessary in the real operation to remove the unrealistic results and to increase the product quality. Therefore, we should evaluate the retrieval results after the quality control. However, as you suggested in the next comments, we need to improve the quality control by using SSA.
The rejected data was 8.2 % in the simulations of the section 4.1, but the rejected data in the analysis of the section 5.1 was 0.07 % for ALM-SW, 0.61% for PPL-SW, 3.12 % for ALM-LW, and 4.37 % for PPL-LW, respectively. In order to prevent the reader from understanding that the rejected data is 8.2% in the operation, we added these values to the section 5.1 of the revised manuscript (lines 455-457).

Figure 1 and corresponding discussion. SSA is a parameter that has values up to 1, and variations in the scale of 0.03-0.05 could change the nature (absorbing) of aerosols. Differences up to 0.5-0.6 are unreasonable and should be filtered out by some QC algorithm (these are SSA values of 0.2 or 0.3). Probably you should implement this plot to a zoomed one in the reasonable range. Still the thickness of the main cloud of data seems to wide to have a robust retrieval, and it doesn't seem to get narrower at higher SZA or AOD (at least from what I can see at this scale). For AOD<0.2 the errors are higher. Differences for shown in table 3 are too high for water soluble aerosols. Probably

an acceptable approach could be to ignore SSA retrievals for very low AOD. Finally, I cannot understand why for the lower uncertainties are in the UV region and not in the visible.

- Thank you. The QC using SSA is a good idea. However, we cannot use it in this study because a deeper investigation is necessary to determine the threshold of SSA.
- We changed the axis range of Fig. 1.
- We agree that the errors of SSA do not become narrower in higher SZA and AOD.
- In general, the error of the retrieved SSA is smaller with higher AOD because the diffuse radiances become more sensitive to SSA (Dubovik et al., 2000). In the UV region, the AOD is high, and the diffuse radiances have the rich information on SSA.

Dubovik, O., Smirnov, A., Holben, B. N., King, M. D., Kaufman, Y. J., Eck, T. F., and Slutsker, I. (2000), Accuracy assessments of aerosol optical properties retrieved from Aerosol Robotic Network (AERONET) Sun and sky radiance measurements, J. Geophys. Res., 105 (D8), 9791– 9806, doi:10.1029/2000JD900040.

L365. After the discussion about the advantage of this method In retrieving bigger particles, I don't understand why this analysis is not up to 30μm.

In the retrieval of the aerosol optical properties, it is most important to decrease the errors of the retrieved VSD around the mode radius. Therefore, we showed only the errors of the retrieved VSD around the mode radius. In the revised manuscript (lines 389-392), we described the errors of the retrieved VSD at the radii larger than the coarse mode radius + one sigma. The errors were more than 100 % for Water-soluble and Biomass burning cases, but that for Dust case was -6+/-57 % for AOD < 0.2, and -9+/-24 % for AOD > 0.2. In the dust case, the error of the retrieved VSD was small.

L390 even for PWV and TO3. I assume this means even for these retrievals. Please restate to be clear.

The sentence was revised (lines 411-413). We mean the retrieved PWV and TO3 are also affected by the aerosol vertical profile.

Figure 8 The caption should be rewritten in a clearer way.

The caption was revised.

Figure 10: It seems that MRIv2 retrieves lower SSA even for high AOD, at all wavelengths above 500nm, compared to both V42 and V5. I strongly suggest narrowing the axis range, because all the info is concentrated in the upper right corner and it cannot

be seen.

Yes, the difference is about 0.05 in the high AOD cases. I revised the figure 10.

Figure 14: No units for both x and y axis.

Thanks, we corrected it.

L 565 giant particles. Since you refer to particles up to 100μm and even retrieve SD up to 30μm, It does not add up to name the 10μm particles as giants.

We deleted "giant" (lines 655, 657).

L569-574 It seems that the fact that AOD was 1/3 compared to 16 August could explain a big part of the inconsistencies with the in situ measurements.

The cause of inconsistency between the retrieval and in-situ measurement is the horizontal heterogeneity of the transported dust and the spatial difference of the aircraft and sky radiometer observations (line 662). The sky radiometer observed the low AOD region of the transported dust.

Figure A2, please express the complex number with same number of significant digits.

Thanks, we corrected it.

[revised manuscript text omitted]